# Analyzing and Mitigating Object Hallucination in Large Vision-Language Models

**Yiyang Zhou**[1*]   **Chenhang Cui**[1*]   **Jaehong Yoon**[1]   **Linjun Zhang**[2]   **Zhun Deng**[3]
**Chelsea Finn**[4]   **Mohit Bansal**[1]   **Huaxiu Yao**[1]
[1]UNC-Chapel Hill, [2]Rutgers University, [3]Columbia University, [4]Stanford University
`zhouyiyangailab@gmail.com, osallymalone@gmail.com, huaxiu@cs.unc.edu`

## Abstract

Large vision-language models (LVLMs) have shown remarkable abilities in understanding visual information with human languages. However, LVLMs still suffer from object hallucination, which is the problem of generating descriptions that include objects that do not actually exist in the images. This can negatively impact many vision-language tasks, such as visual summarization and reasoning. To address this issue, we propose a simple yet powerful algorithm, **LVLM Hallucination Revisor (LURE)**, to post-hoc rectify object hallucination in LVLMs by reconstructing less hallucinatory descriptions. LURE is grounded in a rigorous statistical analysis of the key factors underlying object hallucination, including co-occurrence (the frequent appearance of certain objects alongside others in images), uncertainty (objects with higher uncertainty during LVLM decoding), and object position (hallucination often appears in the later part of the generated text). LURE can also be seamlessly integrated with any LVLMs. We evaluate LURE on six open-source LVLMs and found it outperforms the previous best approach in both general object hallucination evaluation metrics, GPT, and human evaluations. Our data and code are available at `https://github.com/YiyangZhou/LURE`.

## 1 Introduction

Large Vision-Language Models (LVLMs) have made significant progress in understanding real-world images, showing potential towards achieving general artificial intelligence (Liu et al., 2023d; Zhu et al., 2023; Ye et al., 2023; Li et al., 2023a; Maaz et al., 2023; Gong et al., 2023). Although LVLMs have demonstrated their versatility and linguistic fluency, they often suffer from *object hallucination* in their generated text outputs (Wang et al., 2023a; Liu et al., 2023a; Gunjal et al., 2023). Object hallucination refers to the phenomenon of generating inaccurate descriptions for a given image, including non-existent objects or omitting essential features. The issue with hallucinatory text generation in LVLMs is that it can mislead and deceive users in downstream applications that depend on these captions or descriptions, ultimately resulting in a negative impact on various fields that employ LVLMs, including robotics (Mai et al., 2023; Liu et al., 2023b), medical imaging (Wang et al., 2023b; Hu et al., 2023), and human-computer interaction (Olson et al., 1994; Brie et al., 2023).

Early works have attempted to address the problem of object hallucinations in small-scale multimodal pre-trained models by performing either fine-grained alignment across different modalities (Biten et al., 2022) or reducing object co-occurrence patterns with data augmentation (Rohrbach et al., 2018; Kim et al., 2023). However, the auto-regressive architecture of LVLMs differs significantly from small-scale multimodal pre-trained models, making their direct utilization impractical. A few recent works (Li et al., 2023c; Liu et al., 2023a;d) have studied to reduce object hallucinations in LVLMs by enhancing the quality of datasets used for fine-tuning. Yet, acquiring a substantial number of high-quality examples for fine-tuning can be time-consuming and labor-intensive, requiring human expertise and effort. Instead, we aim to propose a lightweight method to post-hoc handle object hallucination by introducing **LURE**: **L**VLM hallc**U**ination **RE**visor.

Concretely, LURE is grounded in a rigorous statistical analysis that elucidates the underlying causalities of object hallucinations in LVLMs. This analysis delves into the relationship between the pre-training data and their corresponding textual responses from LVLMs that exhibit hallucinatory

---

*Equal contribution. Work was done during Yiyang Zhou and Chenhang Cui's remote internship at UNC.

contents (Ordonez et al., 2011; Lin et al., 2014; Changpinyo et al., 2021; Liu et al., 2023d). Both our empirical and theoretical findings reveal that object hallucinations can be attributed to three key factors: co-occurrence, uncertainty, and object position. First, if the training data contains spurious co-occurring patterns between objects, language models may generate outputs based on these learned spurious associations, thus resulting in hallucinatory descriptions. Second, hallucinations occur more frequently on objects characterized by high uncertainty during generation. Lastly, positional factors also play a role, as more object hallucinations tend to appear in the latter portions of the generated description due to the accumulation of misinterpretation.

Based on our statistical analysis, LURE develops a object hallucination revisor. This revisor takes potentially hallucinatory descriptions as input and converts them into accurate ones. To create the revisor, we first generate a hallucinatory dataset using GPT-3.5 by making two modifications to the original correct captions: (1) Insert additional object texts into the description that are likely to co-occur with the objects contained in the initial description. This modification allows LURE to learn to disentangle such co-occurrence patterns effectively; (2) Replace uncertain objects or those at the end of descriptions with a placeholder tag, encouraging the revisor to re-evaluate these objects. In the end, we train our *hallucination revisor* leveraging the acquired hallucinatory dataset. Once trained, the revisor can seamlessly integrate with any LVLM to correct potential hallucinatory descriptions.

Our primary contribution is LURE, a lightweight and compatible post-hoc approach for rectifying object hallucination in LVLMs. This approach is grounded in our rigorous statistical analyses of object hallucinatory phenomena in LVLMs. Our experiments thoroughly evaluate LURE on multiple existing open-source LVLMs. Compared to the best prior method, the results demonstrate that LURE can significantly reduce object hallucination under general object hallucination evaluation metrics (e.g., CHAIR (Rohrbach et al., 2018)), GPT evaluation, and human evaluation.

## 2 WHY DO LARGE VISION-LANGUAGE MODELS EXPERIENCE OBJECT HALLUCINATION?

This section scrutinizes the root causes of object hallucinations in vision-language models via comprehensive statistical analyses from three critical viewpoints: *co-occurrence*, *uncertainty*, and *position*, recognized as the primary factors contributing to object hallucination. We further provide a rigorous theoretical explanation that complements our empirical findings on object hallucinations.

**Notations.** Large Vision-Language Models (LVLMs) typically generate sentences in a free-form and auto-regressive manner, predicting the probability distribution of the next token progressively. In this context, we denote the input as $x$, the correct answer as $y$, and the generated sequence with a length of $N_s$ as $s = \{z_1, \ldots, z_{N_s}\}$. For a given LVLM, the probability of generating $z_i$ as the $i$-th token can be described as $p(z_i|s_{<i}, x)$ (where $1 \leq i \leq N_s$), and $s_{<i}$ refers to the previously generated tokens $\{z_1, \ldots, z_{i-1}\}$. Given a description $s$, we additionally define the complete object set, which is arranged in the order of appearance, as $\mathcal{O}_s = \{o_{s,1}, \ldots, o_{s,n_h+n_r}\}$. Here, $n_h$ and $n_r$ represent the number of hallucinatory and non-hallucinatory objects, respectively.

### 2.1 CO-OCCURRENCE AND SPURIOUS CORRELATION AMONG OBJECTS

In the realm of multi-modal models, "co-occurrence" denotes the frequent appearance of specific objects. When the training data includes spurious co-occurring patterns among objects, language models can generate outputs based on these learned associations. However, these associations may not hold true for test examples, resulting in hallucinatory outputs. For example, "grass" and "sky" frequently co-occur in the training data. The model falsely associates them and tends to generate "grass" and "sky" together even when only "grass" is present in the context.

In order to assess the influence of co-occurrence on object hallucination, we draw inspiration from Biten et al. (2022)and introduce a *Co-occurrence Score* denoted as CoScore. For each image description $s$, the corresponding co-occurrence score $\text{CoScore}_s$ is computed as the summation of co-occurrence degrees across all hallucinatory objects $\{o_{s,1}, \ldots, o_{s,n_h}\}$, which is defined as:

$$\text{CoScore}_s = \sum_{i=1}^{n_h} \sum_{j=1, o_{s,j} \neq o_{s,i}}^{n_r+n_h} \frac{|\mathcal{S}(o_{s,i}) \cap \mathcal{S}(o_{s,j})|}{|\mathcal{S}(o_{s,i})| + |\mathcal{S}(o_{s,j})|}. \tag{1}$$

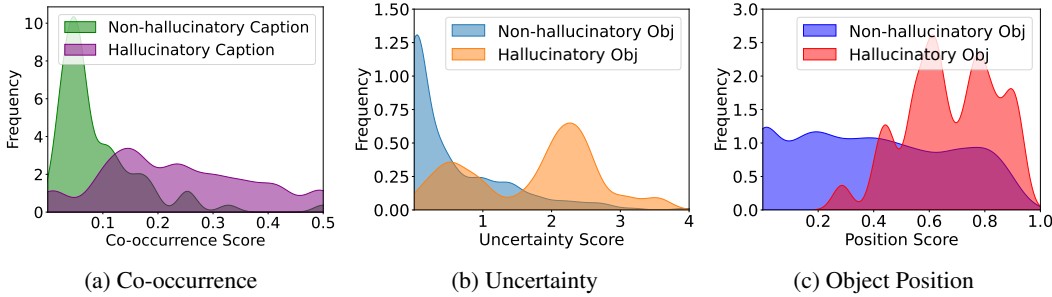

|  (a) Co-occurrence | (b) Uncertainty | (c) Object Position |

Figure 1: Comparison between hallucinatory and non-hallucinatory captions under different factors.

Here, $\mathcal{S}(\cdot)$ denotes the set of all descriptions that mention a specific object, and $|\mathcal{S}(\cdot)|$ represents the cardinality of this set.

Based on the definition of $\mathrm{CoScore}$, we compare the distribution of co-occurrence scores between hallucinatory and non-hallucinatory captions (please refer to Appendix A.1 for our experimental setting), As shown in Figure 1a, hallucinatory captions tend to exhibit higher co-occurrence scores, which suggests a stronger association between object hallucination and co-occurrence.

## 2.2 OBJECT UNCERTAINTY

In language modeling, beam search (Holtzman et al., 2019; Freitag & Al-Onaizan, 2017) is employed to predict words iteratively, introducing inherent uncertainty into the search process (Please refer to illustrative examples in Appendix D.1). This uncertainty is used as a measure of the model's confidence in generating the next token, and can be related to the hallucination problem, as objects with higher uncertainty are more likely to be inaccurate. Here, we aim to quantitatively investigate the potential relationship between the uncertainty associated with objects at each prediction step and the hallucinations.

Concretely, we represent the probability of autoregressive decoding for each object token as $p(o_{s,i}|s_{<k}, x)$, where $k$ denotes the positional index of object $o_{s,i}$. For each object $o_{s,i}$, the corresponding *Uncertainty Score* is defined as:

$$\mathrm{UnScore}_{s,i} = -\log p(o_{s,i}|s_{<i}, x), \tag{2}$$

where a higher value of the uncertainty score indicates greater uncertainty. In Figure 1b, we perform a statistical analysis examining the connection between hallucination and object uncertainty (refer to Appendix A.1 for experimental details). Similar to the analysis of co-occurrence, hallucinatory objects are predominantly observed in the high-uncertainty range, while non-hallucinatory objects are more frequently generated in the certain range.

## 2.3 OBJECT POSITION IN GENERATED DESCRIPTIONS

We also find a significant correlation between the object position in the generated descriptions and hallucination, where dominant hallucinations occur in the latter part of the descriptions. To validate it, we introduce the *Positioning Score* denoted as $\mathrm{PoScore}$ for each object $o_{s,i}$ as follows:

$$\mathrm{PoScore}_{s,i} = \frac{\mathrm{Index}(o_{s,i})}{N_s}, \tag{3}$$

where $\mathrm{Index}(o_{s,i})$ signifies the position index of object $o_{s,i}$ within the entire description.

Based on the definition of $\mathrm{PoScore}$, we conduct a analysis of the positions of hallucination in the descriptions, illustrated in Figure 1c (refer to Appendix A.1 for experimental details and Appendix C.1.1 for more analysis). These findings indicate that high-density areas of hallucinatory objects predominantly appear towards the end of the sequence. This pattern corroborates our observation that object hallucination frequently occurs in the latter segments of generated text. One plausible explanation for this observed trend is rooted in the autoregressive text generation process. In the initial stages, the model closely adheres to the semantic information of its input image, resulting in coherent beginnings. However, as the generation progresses, the accumulation of past hallucinatory information and emerging uncertainties may steer the model off-course, ultimately leading to a more pronounced emergence of object hallucination.

## 2.4 THEORETICAL EXPLANATION

After examining these empirical correlations, we proceed to offer theoretical insights to explain them (all proofs can be found in Appendix B). Specifically, we focus on predicting the $i$-th token, denoted as $z_i$, and introduce a predictive function denoted as $f$. For each object $k$ within a set of objects represented as $[K]$, the function $f_k(s_{<i}, x)$ signifies the predicted score associated with the $k$-th object. Here, $K$ is defined as the total number of objects under consideration, and we use $y_k = 1$ to denote the presence of the $k$-th object in an image and $y_k = -1$ otherwise. Furthermore, we make an assumption that $f_k(s_{<i}, x)$ can be expressed as $\langle \phi_k(s_{<i}, x), \beta_k \rangle$, $\phi_k(s_{<i}, x) \mid y_k \sim N(y_k \cdot \mu_k^*, I_d)$ and $\Pr(y_k = 1) = \Pr(y_k = -1) = 1/2$. For a training set $\mathcal{D}$, the optimizer for the $k$-th class parameter $\beta_k$ trained on $\mathcal{D}$ is defined as: $\hat{\beta}_k = \frac{1}{|\mathcal{D}|} \sum_{(s_{<i}, x, y_{i,k}) \in \mathcal{D}} y_{i,k} \cdot \phi_k(s_{<i}, x)$, where $y_{i,k} \in \{-1, 1\}$ represents whether object $k$ will occur at position $i$. Such a model and optimizer are commonly used in the theoretical analysis of deep learning models (Carmon et al., 2019; Zhang et al., 2022a).

**Co-occurrence.** Based on this definition, we first consider co-occurrence. Without loss of generality, we assume that $K = 2$, and the first and second classes are frequently observed together, i.e., we observe $(\phi_1(s_{<i}, x), \phi_2(s_{<i}, x))$ among a fraction $\rho_0 \in (0, 1)$ of samples when both $y_1$ and $y_2$ are equal to 1. Here, to simplify the autoregressive process while maintaining sequential prediction manner, we consider using $\hat{f}_1 = \langle \phi_1(s_{<i}, x), \hat{\beta}_1 \rangle$ for the prediction of the first object, and in the second prediction, we model the information passed from the first information by $\langle \phi_1(s_{<i}, x), \hat{\beta}_1 \rangle$, and consider $\hat{f}_2 = \langle \phi_1(s_{<i}, x), \hat{\beta}_1 \rangle + \langle \phi_2(s_{<i}, x), \hat{\beta}_2 \rangle$. The model outputs the second object if $\hat{f}_2(s_{<i}, x) > 0$.

Under this setting, we consider two sampling schemes: (1) Each class is sampled according to the original training distribution; (2) Each class is sampled by setting $\rho < \rho_0$. These two sampling schemes result in two subset of samples $\mathcal{D}^{(1)}, \mathcal{D}^{(2)}$ with the same size. Denote the classifiers trained on $\mathcal{D}^{(1)}$ and $\mathcal{D}^{(2)}$ by $\{\hat{f}_k^{(1)}\}_{k \in \{1,2\}}$ and $\{\hat{f}_k^{(2)}\}_{k \in \{1,2\}}$ respectively. Theorem 2.1 reflect reducing co-occurrence issue can lead to smaller test misclassification error $Err(\cdot)$.

**Theorem 2.1** *Suppose $\|\mu_k^*\|^2 \ll d$, $d/|\mathcal{D}^{(k)}| \to \kappa$ for $k \in \{1, 2\}$ and universal constants $\kappa > 0$. We have*
$$Err(\hat{f}_2^{(2)}) \leq Err(\hat{f}_2^{(1)}).$$

**Uncertainty.** We then turn our attention to object uncertainty. Here, we consider the two following sampling schemes: (1) Each class is sampled with equal probability $1/K$; (2) Each class is sampled if the uncertainty score, defined as $-\log(\hat{p}_k)$, is above a certain threshold $\gamma > 0$. Here, $\hat{p}_k$ is calculated as follows: $\hat{p}_k = \frac{1}{|\mathcal{D}^{tr}|} \sum_{(s_{<i}, x, 1)} \sigma(\langle \phi_k(s_{<i}, x), \hat{\beta}_k \rangle)$, where $\mathcal{D}^{tr}$ represents the training set. These two schemes result in two subsets of samples $\mathcal{D}^{(1)}$ and $\mathcal{D}^{(2)}$ with the same size. Given $x$ and $s_{<i}$, we make a prediction about whether the $k$-th object is present in the image using $\hat{f}_k$. Theorem 2.2 illustrates that sampling more certain objects can lead to a reduction in test error.

**Theorem 2.2** *Suppose $\|\mu_k^*\|^2 \ll p$, $d/|\mathcal{D}^{(k)}| \to \kappa$ for $\kappa > 0$ and $k \in [K]$. We will have with probability at least $1 - o(1)$,*
$$\frac{1}{K} \sum_{k=1}^{K} Err(\hat{f}_k^{(2)}) \leq \frac{1}{K} \sum_{k=1}^{K} Err(\hat{f}_k^{(1)}).$$

**Object Position.** The effect of object position on object hallucination is closely tied to error or prediction uncertainty accumulation in autoregressive models. This topic has been extensively studied in time series analysis, and several theoretical models have been established to investigate it (Hannan et al., 1989; Ing, 2007; Ding et al., 2017).

## 3 LVLM HALLUCINATION REVISOR

After thoroughly investigating the root causes of hallucinations, this section formally introduces our remedy, LURE, that mitigates object hallucinations in large vision-language models. Inspired by denoising autoencoders (Vincent et al., 2008), which is designed to reconstruct clean data from corrupted input, we employ a hallucination revisor in our approach that aims to transform potentially LVLM-generated hallucinatory descriptions into accurate ones. The framework of LURE is depicted in Figure 2. In the subsequent sections, we will delve into the training and deployment processes of the hallucination revisor.

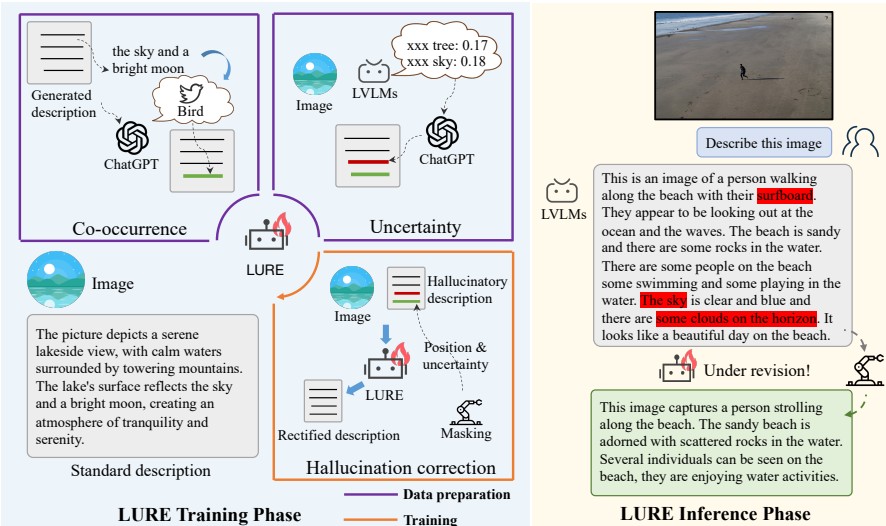

Figure 2: An illustration of LURE Framework: The orange-shaded section shows the training paradigm of LURE, where the black-bordered part represents the hallucinatory data generation phase, including introducing co-occurring objects and replacing either uncertain objects or objects in later positions in the descriptions. The purple-bordered part indicates the revisor training process, with the masking process that can be referenced in Alg. 1. The orange-shaded section illustrates an example in the inference phase of LURE.

## 3.1 TRAINING HALLUCINATION REVISOR

In LURE, to train the hallucination revisor, we first curate a training dataset. Each example in this dataset consists of an image accompanied by a hallucinatory description, with the correct description serving as the output target. A significant challenge encountered during dataset curation lies in the generation of naturally-occurring hallucinatory descriptions. To overcome this challenge, LURE generates hallucinatory descriptions by modifying the accurate descriptions using GPT-3.5. These adjustments are guided by factors related to object hallucination, including co-occurrence, object uncertainty, and object position. In the following, we detail these modifications:

**Introducing Potential Co-Occurrence Objects.** To create a more naturally occurring co-occurrence scenario, rather than relying on counting co-occurrence frequencies from any specific datasets that may contain bias co-occurrence records, LURE leverages GPT-3.5 to deduce and incorporate objects that are most likely to co-occur in the scene into the original description.

**Reconsidering Uncertain Objects & Objects in Later Position in the Descriptions.** Hallucination is more prone to occur in objects with greater uncertainty and objects exist later in the description. In this context, we anticipate that the revisor should place greater emphasis on and reevaluate these objects. To achieve this, we utilize string matching to replace objects with significant uncertainty and those located at the end of the description with the placeholder tag "[IDK]".

Here, to quantify object uncertainty in descriptions, we use the uncertainty values of noun tokens as a proxy. Token uncertainty is expressed as the entropy of each token, denoted as $-\log p(z_i|s_{<i}, x)$. We classify tokens as uncertain objects if their corresponding uncertainty exceeds a threshold $\gamma$, and if they are identified as nouns. Like uncertainty, we determine the later object's position using the condition $\mathrm{Index}(z_i) \geq \eta * \mathrm{Length}(s)$ and the threshold $\eta$. This approach enables the model to reassess and either replace "[IDK]" with a more appropriate object based on the image or remove it entirely. Using these modification strategies, for every accurate description, we provide GPT-3.5 with a list of potential co-occurrence objects, and a list of uncertain objects.

We then prompt GPT-3.5 to generate the corresponding hallucinatory description using the prompts listed in Appendix A.3. Finally, we leverage the constructed hallucination dataset to fine-tune a LVLM and use it as revisor. Some cases of hallucinatory descriptions are in Appendix D.2. The training pipeline is illustrated in Alg. 1.

---

**Algorithm 1** Training LVLM Hallucination Revisor in LURE

---

**Require:** training image set $\mathcal{X}$; groundtruth descriptions $\mathcal{Y}$; LVLM $\mathcal{M}(\cdot)$; uncertainty threshold $\gamma$; hallucination revisor $\mathcal{R}_\theta(\cdot)$ with parameters $\theta$; position threshold $\eta$

1: Use GPT-3.5 to construct hallucinatory description set $\mathcal{H}_{old}$ (see Appendix A.3 for more details)
2: Initialize the revisor's parameter $\theta$ and an empty set $\mathcal{H}_{new} \leftarrow \{\}$
3: **while** not converged **do**
4:     **for** each image $x \in \mathcal{X}$ and the correpsonding hallucinatory description $h \in \mathcal{H}_{old}$ **do**
5:         Generate description $s = \mathcal{M}(x)$ with object set $\mathcal{O}_s$
6:         **for** object $o_{s,i} \in \mathcal{O}_s$ **do**
7:             **if** $o_{s,i}$ **in** $h$ and $-\log p(o_{s,i}|\mathcal{M}, x) \geq \gamma$ **then**
8:                 Add placeholder tag "[IDK]" to $h$, i.e., $h \leftarrow \text{Mask}(h, o_{s,i})$
9:             **if** $o_{s,i}$ **in** $h$ and $\text{Index}(o_{s,i}) \geq \eta * \text{Length}(h)$ **then**
10:                Add placeholder tag "[IDK]" to $h$, i.e., $h \leftarrow \text{Mask}(h, o_{s,i})$
            Put $h$ into $H_{new}$
11:     Update parameter $\theta$ with autoregressive loss $\mathcal{L}(\mathcal{R}_\theta(H_{new}), \mathcal{Y})$

---

## 3.2 DEPLOYING HALLUCINATION REVISOR

In the inference stage, the trained revisor is employed to rectify the generated descriptions. Specifically, similar to the process of constructing hallucinated descriptions during the training phase, in the testing phase, we similarly integrate the placeholder tag "[IDK]" into the generated descriptions. This integration serves the purpose of enforcing the Revisor to reevaluate objects exhibiting high uncertainty or appearing later in the generated text. The inference pipeline is detailed in Alg. 2.

---

**Algorithm 2** Inference Pipline of LURE

---

**Require:** test image $x_t$; LVLM $\mathcal{M}(\cdot)$; trained hallucination revisor $\mathcal{R}_\theta^*(\cdot)$; uncertainty threshold $\gamma$, position threshold $\eta$

1: Generate description $s_t = \mathcal{M}(x_t)$ with object set $\mathcal{O}_{s_t}$
2: **for** object $o_{s_t,i} \in \mathcal{O}_{s_t}$ **do**
3:     **if** $-\log p(\text{object}|\mathcal{M}, x) \geq \gamma$ **then**
4:         Add placeholder tag "[IDK]" to $s_t$, i.e., $s_t \leftarrow \text{Mask}(s_t, o_{s_t,i})$
5:     **if** $\text{Index}(o_{s_t,i}) \geq \eta * \text{Length}(s_t)$ **then**
6:         Add placeholder tag "[IDK]" to $s_t$, i.e., $s_t \leftarrow \text{Mask}(s_t, o_{s_t,i})$
7: **return** $\mathcal{R}_\theta^*(s_t)$

---

## 4 EXPERIMENTS

In this section, we evaluate the performance of LURE aiming to answer the following questions: (1) Can LURE effectively reduce object hallucination in LVLMs compared to other baselines? (2) Can the key factors we've identified related to hallucinations in LVLMs benefit the training process of the revisor? (3) Is LURE sensitive to the revisor's backbone?

**Datasets.** MSCOCO (Lin et al., 2014) is a comprehensive dataset used for image recognition, segmentation, and captioning. It comprises over 300,000 images spanning more than 80 object categories, each with detailed annotations. Following (Li et al., 2023d; Liu et al., 2023a), we selected 5,000 unique images from the COCO 2014 training dataset to evaluate performance. To train the hallucination revisor, we randomly selected 5000 image-text pairs from LLaVA-150k (Liu et al., 2023c), ensuring that these images were different from the ones used in testing. In addition, we also evaluate the performance on other datasets, as discussed in Appendices B.4 and B.5.

**Evaluation Metric.** Caption Hallucination Assessment with Image Relevance (CHAIR) (Rohrbach et al., 2018) is a widely-used metric for evaluating object hallucination in image captioning tasks. CHAIR assesses the quality of image captions by comparing them to the ground truth objects present in the corresponding images. It calculates the proportion of objects mentioned in the caption that are not actually present in the image. There are two common variants of CHAIR: $\text{CHAIR}_I$ and $\text{CHAIR}_S$. Both variants evaluate the degree of object hallucination, but at different levels: the object instance level and the sentence level, respectively. The two variants are formulated as follows:

$$\text{CHAIR}_I = \frac{|\{\text{hallucinated objects}\}|}{|\{\text{all mentioned objects}\}|}, \quad \text{CHAIR}_S = \frac{|\{\text{captions with hallucinated objects}\}|}{|\{\text{all captions}\}|}. \quad (4)$$

**Baselines.** The comparison methods include: *Original*, which directly use the generated descriptions from LVLMs; *Teacher* (Saha et al., 2023), which leverages blip2 (Li et al., 2023b) to generate short image descriptions and employs them as contextual guidance for generating long-form descriptions; *Chain-of-Thought (CoT)* (Wei et al., 2022), which involves the model initially listing objects and subsequently describing the image; *Greedy-Decoding*, a method that abstains from using a sampling strategy and aims to make the model output the most certain tokens; *GPT-Ensemble*, which initially employs GPT-3.5 to aggregate the commonly generated descriptions from multiple LVLMs, excluding the one under evaluation. Subsequently, GPT-3.5 utilizes these summarized common descriptions as guidance to rewrite the originally generated description from the evaluated model; *GPT-Teacher*, where GPT-3.5 is tasked with rewriting the original long-form description based on the blip2 generated short descriptions. Detailed descriptions about baselines are in Appendix A.4.

**Evaluated LVLMs.** We performed experiments utilizing six of the most recent LVLMs, with their corresponding language models specified in parentheses: MiniGPT-4 (Vicuna 13B) (Zhu et al., 2023), LLaVa (LLaMA 13B) (Liu et al., 2023d), MMGPT (LLaMA 7B) (Gong et al., 2023), LLaMA-Adapter (LLaMA 7B) (Zhang et al., 2023b), mPLUG-Owl (LLaMA 7B) (Ye et al., 2023), and InstructBLIP (Vicuna 7B) (Dai et al., 2023).

**Hyperparameter Settings.** Unless specified, all experiments in the paper are using MiniGPT-4 as the backbone of the revisor, along with the training parameter settings provided in Appendix A.2. All hyperparameters are selected via cross-validation.

## 4.1 EVALUATION STRATEGIES AND RESULTS

**Automated Object Hallucination Evaluation.** We follow the guidelines presented in (Rohrbach et al., 2018) to perform an automated calculation of CHAIR metrics for the MSCOCO dataset, where 80 objects are involved in this automated evaluation process. In addition, we extend our evaluation to include other widely used metrics such as BLEU and CLIP score, which are commonly adopted in assessing the quality of image captioning. Detailed descriptions and results for these additional metrics can be found in Appendix B.3.

**Human and GPT Evaluations.** Although automated evaluation strategies are efficient, they cannot encompass all objects present in the evaluated images. To overcome this limitation, we conducted a comprehensive human evaluation involving several native speakers. Please refer to Appendix A.5 for the evaluation interface. In this human evaluation, participants are assigned the task of annotating hallucinatory objects and we rank different methods based on human feedback. In addition to human evaluation, inspired from (Zheng et al., 2023), we also prompt GPT-3.5 to compare different descriptions. In this GPT evaluation, we provide the annotated information, including detection boxes and captions, and anticipate that GPT-3.5 can provide an ranking for the descriptions from various methods. For GPT evaluation, we use the prompts referenced in Table 9 in the Appendix.

**Results.** In Table 1 and Table 2, we report the results of automated evaluations and human and GPT evaluations under different LVLMs, respectively (see more analysis about the effectiveness of LURE on Appendices C.2 and C.1.3). Here, taking cost into account, we only compare LURE with the four strongest methods in human and GPT evaluations. Although Teacher, CoT, and GPT-Teacher can improve the performance compared to the original descriptions in most cases, LURE significantly enhances performance over these strong baselines, which effectively reduces object hallucination in generated descriptions. One potential reason for this is that all of these baselines experience error propagation to some extent. For instance, CoT's linear guidance can lead to errors if the object listing step is incorrect. In contrast, LURE directly corrects hallucinatory descriptions using guidance from potential factors that can trigger hallucinations.

## 4.2 ANALYSIS OF LURE

**Are the Performance Gains of LURE from Using Constructed Hallucination Datasets?** To verify that the performance gains of our method are not from using additional data to train the revisor, we fine-tuned the original LVLMs with the additional dataset. The results on MiniGPT-4 are shown in Table 3, where "Original" represents the descriptions

Table 3: Compared LURE to fine-tuning method using the training data of revisor.

| Model | CHAIR$_S \downarrow$ | CHAIR$_I \downarrow$ |
|---|---|---|
| Original | 26.8 | 7.3 |
| FT (add'l data) | 31.0 | 7.2 |
| **LURE (Ours)** | **19.7** | **4.9** |

Table 1: Automated hallucination evaluation is performed under six LVLMs using CHAIR$_S$ ($C_S$) and CHAIR$_I$ ($C_I$), where smaller values indicate less object hallucination. For additional metrics, please refer to Appendix B.3.

| | MiniGPT-4 | | LLaVa | | MMGPT | | LLaMA-Adapter | | mPLUG-Owl | | InstructBLIP | |
| | $C_S \downarrow$ | $C_I \downarrow$ | $C_S \downarrow$ | $C_I \downarrow$ | $C_S \downarrow$ | $C_I \downarrow$ | $C_S \downarrow$ | $C_I \downarrow$ | $C_S \downarrow$ | $C_I \downarrow$ | $C_S \downarrow$ | $C_I \downarrow$ |
|---|---|---|---|---|---|---|---|---|---|---|---|---|
| Original | 26.8 | 7.3 | 54.0 | 11.3 | 56.6 | 11.0 | 58.8 | 13.7 | 71.2 | 16.5 | 40.0 | 8.2 |
| Teacher | 24.0 | 5.7 | 49.9 | 9.3 | 53.4 | 7.5 | 40.8 | 9.4 | 62.4 | 13.0 | 36.4 | 7.5 |
| CoT | 31.6 | 9.4 | 47.6 | 9.0 | 48.8 | 17.5 | 43.3 | 9.4 | 56.9 | 13.4 | 35.7 | 7.8 |
| Greedy-Decoding | 25.1 | 6.6 | 50.9 | 10.0 | 50.6 | 8.4 | 55.9 | 13.7 | 55.1 | 12.8 | 35.5 | 7.8 |
| GPT-Ensemble | 41.0 | 10.6 | 43.0 | 10.7 | 51.0 | 11.1 | 47.1 | 13.0 | 52.0 | 15.2 | 51.0 | 13.0 |
| GPT-Teacher | 25.3 | 7.6 | 38.0 | 7.8 | 26.7 | 9.3 | 49.0 | 12.4 | 22.0 | 9.0 | 32.0 | 7.8 |
| **LURE (ours)** | **19.7** | **4.9** | **27.1** | **6.4** | **22.2** | **5.6** | **35.3** | **9.1** | **18.8** | **5.4** | **21.0** | **5.1** |

Table 2: We conducted evaluations for description ranking, comparing the four strongest baselines in both human ('H') and GPT ('G') evaluations. Metrics represent the average rankings within the top 1-5 positions, with lower rankings indicating less hallucination.

| | MiniGPT-4 | | LLaVa | | MMGPT | | LLaMA-Adapter | | mPLUG-Owl | | InstructBLIP | |
| | G$\downarrow$ | H$\downarrow$ | G$\downarrow$ | H$\downarrow$ | G$\downarrow$ | H$\downarrow$ | G$\downarrow$ | H$\downarrow$ | G$\downarrow$ | H$\downarrow$ | G$\downarrow$ | H$\downarrow$ |
|---|---|---|---|---|---|---|---|---|---|---|---|---|
| Original | 3.97 | 3.10 | 4.55 | 4.62 | 3.66 | 3.25 | 4.79 | 4.45 | 4.25 | 3.98 | 4.29 | 4.77 |
| Teacher | 3.36 | 3.83 | 3.30 | 3.07 | 3.09 | 3.20 | 3.00 | 3.13 | 3.25 | 3.66 | 3.34 | 3.53 |
| CoT | 2.44 | 2.83 | 3.05 | 2.52 | 4.38 | 4.07 | 2.63 | 2.10 | 3.75 | 3.13 | 2.78 | 2.21 |
| GPT-Teacher | 3.56 | 3.28 | 2.45 | 2.96 | 2.16 | 2.90 | 2.68 | 3.24 | 2.50 | 2.44 | 3.12 | 2.56 |
| **LURE (ours)** | **1.67** | **1.96** | **1.65** | **1.83** | **1.61** | **1.58** | **1.90** | **2.08** | **1.25** | **1.79** | **1.47** | **1.93** |

of MiniGPT-4. According to Table 3, LURE outperforms the fine-tuned LVLMs, which indicates that our method indeed reduces object hallucination by post-hoc rectifying potential hallucinatory descriptions rather than using additional data.

**Ablation Study – Do the Hallucination Factors Contribute Performance Gains?** To demonstrate the impact of considering co-occurrence, uncertainty, and object position in reducing hallucination, we conducted ablation experiments and report the results in Table 4, where "Original" represents the descriptions of MiniGPT-4. In the ablation experiments, we trained and deployed the revisor without each of the three factors, one at a time. The results show that all three factors contribute to training a strong hallucination revisor to reduce object hallucination. Furthermore, we have also conducted an analysis of the changes in these three factors before and after applying the revisor, as presented in Appendix C.1.1. This analysis demonstrates that LURE can effectively reduce instances of hallucination caused by these factors.

Table 4: Ablation studies on three hallucination factors.

| Model | CHAIR$_S \downarrow$ | CHAIR$_I \downarrow$ |
|---|---|---|
| Original | 26.8 | 7.3 |
| w/o Co-occurrence | 22.6 | 4.9 |
| w/o Uncertainty | 21.2 | 5.4 |
| w/o Position | 22.3 | 5.8 |
| **LURE (Ours)** | **19.7** | **4.9** |

**Robustness Analysis of the Hallucination Revisor.** We further analyze the robustness of the revisor with respect to different backbones. Specifically, we trained the revisor on the same dataset using different backbones: MiniGPT-4, LLaMA-adapter, and mPLUG-Owl. The results are reported in Table 5, where "Original" represents the descriptions of MiniGPT-4. We can observe that despite the varying performance of each backbone, LURE consistently improve the performance compared to the original description, which further indicate the effectiveness of LURE. Additionally, we analyze the results of LURE with respect to various uncertainty thresholds in Appendix C.1.3. The findings demonstrate that LURE exhibits strong performance across a wide range of uncertainty thresholds.

Table 5: Performance under different hallucination revisor backbones.

| Backbone | CHAIR$_S \downarrow$ | CHAIR$_I \downarrow$ |
|---|---|---|
| Original | 26.8 | 7.3 |
| MiniGPT-4 | 19.7 | 4.9 |
| LLaMA-adapter | 21.3 | 5.2 |
| mPLUG-Owl | 22.1 | 5.4 |

**Case Analysis.** We select several strong baselines and presented a case with rectified descriptions in Figure 3. Compared with other approaches, LURE excels in providing a more accurate image

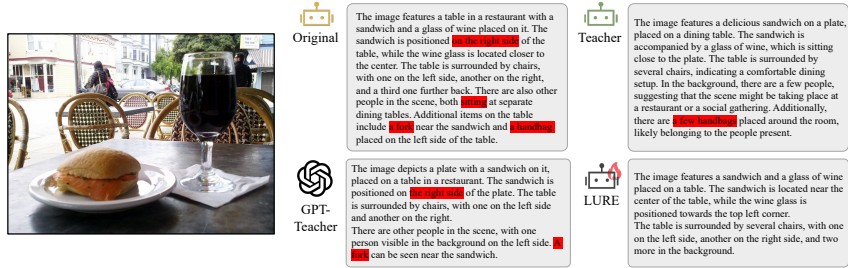

Figure 3: A case study comparing the levels of hallucination among various baselines.

description. In the case, LURE accurately depicts the primary elements (e.g., sandwich, chair, plate) while avoiding hallucinatory objects like the fork and handbag. Although other baselines partially reduce hallucination, they still exhibit object hallucinations in their descriptions. Additionally, we also mitigate logical errors to some extent, including object orientation and actions. Further case analyses can be found in Appendices D.3 and D.4.

## 5 RELATED WORK

**Vision-Language Models.** Vision-language pre-trained models, as exemplified by (Li et al., 2021; Zeng et al., 2021), demonstrate substantial capabilities in modeling interactions between visual and textual information, especially when fine-tuned for specific tasks. Recently, autoregressive large-scale language models (LLMs) (Brown et al., 2020; Chowdhery et al., 2022; Touvron et al., 2023; Zhang et al., 2022b; Chiang et al., 2023; Taori et al., 2023) have ushered in a new era of vision-language models. These models, known as LVLMs, integrate LLMs with visual modality and show-case impressive visual understanding through end-to-end training techniques that directly decode visual and text tokens in a unified manner (Liu et al., 2023d; Zhu et al., 2023; Ye et al., 2023; Li et al., 2023a). However, similar to VLMs, LVLMs also face the challenge of object hallucination (Wang et al., 2023a; Rohrbach et al., 2018). This form of object hallucination is more pronounced and widespread in the long-form descriptions produced by LVLMs compared to the shorter descriptions generated by VLMs (Zhang et al., 2023a).

**Hallucination in VLMs and LVLMs.** In VLMs, hallucination typically refers to scenarios where the generated descriptions contain information that does not exist in the visual modality (Rohrbach et al., 2018; Biten et al., 2022; Wang et al., 2023a). Addressing object hallucination in VLMs is primarily achieved through techniques such as fine-grained contrastive learning (Zeng et al., 2021), ROI feature fusion (Biten et al., 2022), and eliminating co-occurrence patterns through data augmentation (Kim et al., 2023). However, the training paradigms between traditional VLMs and recent LVLMs differ, and the new autoregressive training paradigm in LVLMs makes it challenging to directly apply hallucination mitigation methods used in VLMs to LVLMs. Recent research has begun to address the issue of object hallucination in LVLMs, including hallucination evaluation and detection (Wang et al., 2023a; Liu et al., 2023a; Li et al., 2023d), as well as the construction of higher-quality datasets for fine-tuning (Gunjal et al., 2023; Li et al., 2023c; Liu et al., 2023a;d). Nevertheless, acquiring a substantial number of high-quality examples can be time-consuming and labor-intensive. Instead, grounded in statistical analysis of hallucination, we propose a conceptually different approach, LURE, to post-hoc rectify object hallucination. We have already demonstrated its effectiveness in reducing hallucination and its compatibility with various LVLMs.

## 6 CONCLUSION

In this paper, our objective is to address the challenge of object hallucination in LVLMs. We introduce a lightweight post-hoc method, named LVLM Hallucination Revisor (LURE), designed to rectify object hallucination in the generated descriptions produced by LVLMs. LURE is grounded in three key factors known to contribute to object hallucination: co-occurrence, uncertainty, and object position. These factors have been demonstrated to induce hallucination both empirically and theoretically. Our experiments, conducted on six open-source LVLMs, demonstrate the effectiveness of LURE in mitigating object hallucination in LVLM-generated descriptions.

## REPRODUCIBILITY STATEMENT

For our theoretical results, we present complete proofs for all our claims in Appendix B and engage in a thorough discussion of the assumptions. As for our empirical results, we delve into the details of the experimental setup, introduce additional metrics, and provide a comprehensive overview of baseline details, all of which can be found in Appendices A and B.3. Additionally, in Appendix D, we offer detailed case demonstrations and comparisons. Furthermore, we include template prompts used during these analytical processes within the A.3 and A.4. It is worth noting that we are committed to open-sourcing the code related to our research after publication.

## ACKNOWLEDGEMENT

This work was partially supported by Juniper, NetworksVolkswagen, ONR grant N00014-22-1-2621, NSF-AI Engage Institute DRL-2112635, DARPA ECOLE Program No. HR00112390060, and DARPA Machine Commonsense (MCS) Grant N66001-19-2-4031. We also thank Center for AI Safety and Google Cloud Research Credits program for supporting our computing needs. The views contained in this article are those of the authors and not of the funding agency.

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

## A  EXPERIMENTAL DETAILS

### A.1  EXPERIMENTAL SETTING FOR THE HALLUCINATION ANALYSIS

**Experimental Setting for Co-occurrence Analysis.** The objects in this experiment are based on the 80 object labels annotated in (Rohrbach et al., 2018) from the COCO dataset, and the image descriptions are generated by MiniGPT-4 based on inference results from 5000 images in the COCO 2014 train dataset.

**Experimental Setting for the Uncertainty Analysis.** Because uncertainty and position analysis are relatively independent from co-occurrence, in order to avoid conducting statistical analysis on the training set distribution, the statistical data for uncertainty analysis is derived from MiniGPT-4's descriptions of 200 images from the COCO 2014 test dataset. The computation of uncertainty is performed using $-\log p(z_i|s_{<i}, x)$.

**Experimental Setting for the Analysis of Position of Hallucinated Objects.** Similar to the uncertainty analysis, we used the manually annotated descriptions of MiniGPT-4 for 200 images from the COCO 2014 test dataset, due to the need for precise positioning.

## A.2 TRAINING SETTINGS FOR REVISOR

The overall revisor training setting is similar to MiniGPT-4. Here, we only need one A100 80G GPU for training, which takes approximately 10 minutes. We present hyperparameter settings of the LURE during the training phase, as shown in Table 6.

Table 6: Training hyperparameters.

| Hyperparameters | |
|---|---|
| Training steps | 410 |
| Warmup steps | 50 |
| Max length | 512 |
| Batch size of multi-modal instruction data | 12 |
| Optimizer | AdamW |
| Learning rate | 3e-5 |
| Learning rate decay | Cosine |
| AdamW $\epsilon$ | 1e-6 |
| AdamW $\beta$ | (0.9, 0.999) |
| Weight decay | 0.05 |

## A.3 PROMPTS FOR TRAINING DATASET

We leverage the in-context few-shot learning capability of GPT-3.5 to generate hallucinatory data automatically for revising. Initially, we prompt GPT-3.5 to provide a list of objects that are highly likely to co-occur with the objects mentioned in the given description. Next, we use LVLMs (such as MiniGPT-4) to generate descriptions for the training set of 5000 images. During this process, we will save nouns with $-\log p(z_i|s_{<i}, x)$ greater than the uncertain threshold $\gamma$ in the decoding process to the list of uncertain objects corresponding to each image. Subsequently, we direct the model to take the original description and incorporate a randomly chosen word from the "co-occur objects" list, as well as another randomly chosen word from the "uncertain objects" list, into it. Detailed prompts are listed in Table 7 and a few examples are presented in Table 19.

## A.4 DETAILS ABOUT BASELINE

In this section, we will provide a detailed explanation of the settings used for the baseline in Table 1, including some parameter settings and prompt configurations. The detailed prompt for baselines can be seen in Table 8.

- **Teacher:** The "Teacher" approach involves generating short descriptions for the images via blip2 (Li et al., 2023b) and using them as context to guide the model in generating descriptions. By providing these descriptions as additional information, the model can benefit from the guidance and produce more accurate or relevant descriptions.

- **CoT:** The "CoT" method asks the model to first list the objects it identifies in the image and then describe the image based on those objects. It draws inspiration from the concept of chain of thought (Wei et al., 2022) and aims to guide the model in generating accurate descriptions by focusing on object recognition.

- **Greedy-Decoding:** The difference between the "Greedy-Decoding" strategy and the "Original" strategy is that in the "Greedy-Decoding" strategy, the model uses greedy decoding instead of sampling during the generation of image descriptions to produce the most deterministic output. This approach is used to explore the potential connection between the generation of illusions and the use of sampling.

- **GPT-Ensemble:** In "GPT-Ensemble," we utilize GPT-3.5 to summarize the common elements in the descriptions generated by multiple LVLMs, excluding the one being evaluated. Subsequently, we employ GPT-3.5 to rewrite the description of the evaluated LVLM, using the identified common elements from the descriptions of the other models to correct any dissimilar parts in the evaluated model's description.

Table 7: The prompt for the GPT-3.5 API to generate the required hallucination dataset. "Instruction 1" is used to ask ChatGPT to provide a list of co-occurring objects based on the description, while "Instruction 2" is used to integrate the objects obtained from the co-occurring object list and the objects from the list of uncertain objects into the given description.

---

**Instruction 1:**

List three other objects that you think are most likely to appear with the objects in the scene described below:

{description}

Output in strict accordance with the following format:

Object one

Object two

Object three

---

**Instruction 2:**

Input caption: {description}

co_objects list: {co_objects list}

uncertain_objets list: {uncertain_objets list}

Select one object from "co_objects list" and "uncertain_objects list" respectively and add it to "Input caption" to get "Output caption". (Try not to change the format)

Output caption:

---

- **GPT-Teacher:** "GPT-Teacher" represents the process of providing the GPT-3.5 API with contextual references and descriptions from the model's output, allowing it to revise the inaccurate description generated by the model into a more accurate version based on the contextual information.

### A.5 DETAILS ABOUT MANUAL ANNOTATION EVALUATIONS

The manual evaluation annotation interface provides a user-friendly interface for performing manual annotations and capturing evaluation feedback. The interface is hosted on the Amazon Web Services (AWS) platform, which offers scalability, reliability, and security for handling annotation tasks. As shown in Figure 4, we annotated all objects and hallucinated objects in the descriptions based on the images. We then provided a binary label (0/1) to indicate whether each description contained hallucinations. Based on the fine-grained annotation results, similar to GPT evaluation, we sorted the results from different baselines.

## B DETAILED PROOF

### B.1 PROOF OF THEOREM 2.1

Let us denote $N = |\mathcal{D}^{(1)}| = |\mathcal{D}^{(2)}|$. For the detection rule of the first object, we have

$$\hat{\beta}_k^{(1)} = \frac{1}{|\mathcal{D}^{(1)}|} \sum_{(s_{<i}, x, y_{i,k}) \in \mathcal{D}^{(1)}} y_{i,k} \cdot \phi_k(s_{<i}, x).$$

As $\phi_k(s_{<i}, x) \mid y_{i,k} \sim N(y_{i,k} \cdot \mu_k^*, I)$, we write

$$y_{i,k} \cdot \phi_k(s_{<i}, x) = \mu_k^* + \epsilon_{i,k}.$$

Now, suppose among all samples, a fraction $\rho_0 \in (0, 1)$ of samples have both $y_1$ and $y_2$ are equal to 1. We can then write

$$(\hat{\beta}_1^{(1)}, \hat{\beta}_2^{(1)}) = (\rho_0 \mu_1^* + \frac{1}{N} \sum_{i=1}^{\rho_0 \cdot N} \epsilon_{i,1}, \rho_0 \mu_2^* + \frac{1}{N} \sum_{i=1}^{\rho_0 \cdot N} \epsilon_{i,2}).$$

Table 8: Prompts for baselines.

**Teacher:**

Reference caption:

{blip2 caption}

Please refer to reference caption and describe this picture:

**CoT:**

Human:

Please list the main objects in the picture and strictly follow the following format:

{object1, object2, object3......}

AI:

{objects list}

Human:

Describe this image

AI:

{description}

**GPT-Ensemble:**

Reference captions 1:{description of model 1}

Reference captions 2:{description of model 2}

Reference captions 3:{description of model 3}

Reference captions 4:{description of model 4}

Reference captions 5:{description of model 5}

Original Description:{description}

Synthesizing the commonalities of Reference captions 1-5, and then removing the parts in the Original Description that do not align with the commonalities, while preserving the original format. Answer:

**GPT-Teacher:**

Reference caption:

{blip2 caption}

Original description:

{description}

Rewrite the original description to align it with the reference caption, delete some objects that you think are hallucinations, and keep the original format. Answer:

Use $\Phi(\cdot)$ to denote the cumulative distribution function of a standard normal distribution. Then for the prediction function $\hat{f}_2 = \langle \phi_1(s_{<i}, x), \hat{\beta}_1^{(1)} \rangle + \langle \phi_2(s_{<i}, x), \hat{\beta}_2^{(1)} \rangle$, we have

$$
\begin{aligned}
Err(\hat{f}_2^{(1)}) &= \frac{1}{2}\mathbb{P}(\langle \phi_1(s_{<i}, x), \hat{\beta}_1^{(1)} \rangle + \langle \phi_2(s_{<i}, x), \hat{\beta}_2^{(1)} \rangle < 0 \mid y = 1) \\
&+ \frac{1}{2}\mathbb{P}(\langle \phi_1(s_{<i}, x), \hat{\beta}_1^{(1)} \rangle + \langle \phi_2(s_{<i}, x), \hat{\beta}_2^{(1)} \rangle > 0 \mid y = -1) \\
&= \Phi(-\frac{\langle \mu_1^*, \hat{\beta}_1 \rangle + \langle \beta_2, \hat{\mu}_2^* \rangle}{\sqrt{\|\hat{\beta}_1\|^2 + \|\hat{\beta}_2\|^2}}) \\
&= \Phi(-\frac{\rho_0\|\mu_1^*\|^2 + \rho_0\|\mu_2^*\|^2}{\sqrt{\rho_0^2\|\mu_1^*\|^2 + \rho_0^2\|\mu_2^*\|^2 + \frac{\rho_0 \cdot d}{N} + \frac{\rho_0 \cdot d}{N}}}) + o(1).
\end{aligned}
$$

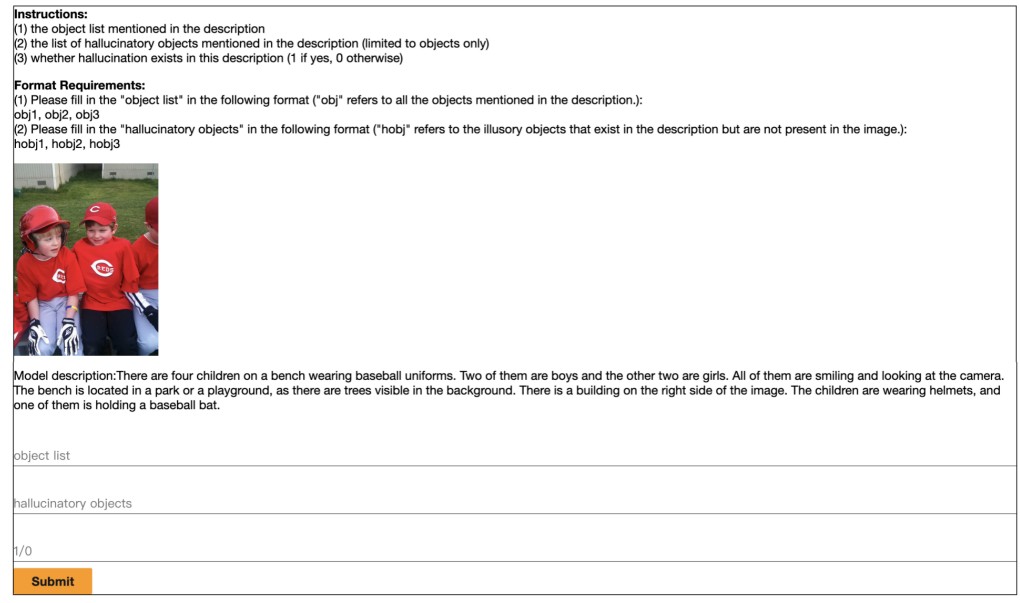

Figure 4: Human evaluation annotation interface.

Table 9: The prompt for ChatGPT3.5 evaluation.

**Instruction:**

Suppose you are a hallucination annotator who judges the degree of hallucination based on objects, and you have the following image information. Reference captions:{five captions from COCO}

Bounding box:{bounding boxes}

Please just provide the ranks for the below descriptions without any explanation, where the caption ranks first with the most hallucinations. The output format: [caption_x,...]

Descriptions:

caption_1: {description_1}

caption_2: {description_2}

caption_3: {description_3}

caption_4: {description_4}

caption_5: {description_5}

Output:

Similarly, we have

$$Err(\hat{f}_2^{(2)}) = \Phi\left(-\frac{\rho\|\mu_1^*\|^2 + \rho\|\mu_2^*\|^2}{\sqrt{\rho^2\|\mu_1^*\|^2 + \rho^2\|\mu_2^*\|^2 + \frac{\rho \cdot d}{N} + \frac{\rho \cdot d}{N}}}\right) + o(1).$$

As $\Phi\left(-\frac{\rho\|\mu_1^*\|^2 + \rho\|\mu_2^*\|^2}{\sqrt{\rho^2\|\mu_1^*\|^2 + \rho^2\|\mu_2^*\|^2 + \frac{\rho \cdot d}{N} + \frac{\rho \cdot d}{N}}}\right)$ is monotonically increasing with $\rho$, we complete the proof.

## B.2 Proof of Theorem 2.2

We first analyze the uncertainty score. In fact, we have

$$
\begin{aligned}
\hat{p}_k =& \frac{1}{|\mathcal{D}^{tr}|} \sum_{(s_{<i}, x, 1)} \sigma(\langle \phi_k(s_{<i}, x), \hat{\beta}_k \rangle) \\
=& \mathbb{E}[\sigma(\langle \phi_k(s_{<i}, x), \hat{\beta}_k \rangle)] + o_P(1) \\
=& \mathbb{E}[\frac{1}{1 + \exp(\|\mu_k^*\|^2 + \|\mu_k^*\| \cdot Z)}] + o_P(1),
\end{aligned}
$$

where $Z \sim N(0, 1)$ is the standard normal random variable.

Therefore, $\hat{p}_k$ decreases when $\|\beta_k\|$ increases. Choosing samples with small $\hat{p}_k$ (i.e., large $-\log(\hat{p}_k)$) correspond to larger sample sizes for the classes with larger $\|\mu_k^*\|$.

Then we analyze the misclassification error. For $\hat{f}_k = sgn(\langle \phi(s_{<i}, x), \hat{\beta}_k \rangle)$, we have

$$
\begin{aligned}
Err(\hat{f}_k) = \mathbb{P}(sgn(\langle \phi(s_{<i}, x), \hat{\beta}_k \rangle) \neq y) =& \frac{1}{2} \mathbb{P}(\langle \phi(s_{<i}, x), \hat{\beta}_k \rangle < 0 \mid y = 1) \\
& + \frac{1}{2} \mathbb{P}(\langle \phi(s_{<i}, x), \hat{\beta}_k \rangle > 0 \mid y = -1)
\end{aligned}
$$

As $\phi_k(s_{<i}, x) \mid y \sim N(y_k \cdot \mu_k^*, I_d)$, we have

$$
\mathbb{P}(\langle \phi_k(s_{<i}, x), \hat{\beta}_k \rangle < 0 \mid y = 1) = \mathbb{P}(\langle \phi(s_{<i}, x), \hat{\beta}_k \rangle > 0 \mid y = -1) = \Phi(-\frac{\langle \mu_k^*, \hat{\beta}_k \rangle}{\|\hat{\beta}_k\|}).
$$

As $\hat{\beta}_k = \mu_k^* + \frac{1}{n_k} \sum_{i=1}^{n_k} \epsilon_i := \mu_k^* + \frac{1}{\sqrt{n_k}} Z$, we have

$$
\frac{\langle \mu_k^*, \hat{\beta}_k \rangle}{\|\hat{\beta}_k\|} = \frac{\|\beta_k\|^2 + \frac{1}{\sqrt{n_k}} \langle \mu_k^*, Z \rangle}{\sqrt{\|\mu_k^*\|^2 + \frac{2}{\sqrt{n_k}} \langle \mu_k^*, Z \rangle + \frac{1}{n_k} \|Z\|^2}}.
$$

As we assume $\|\mu_k^*\|^2 \ll d$, we have

$$
\frac{\langle \mu_k^*, \hat{\beta}_k \rangle}{\|\hat{\beta}_k\|} = \frac{\|\mu_k^*\|^2}{\sqrt{\|\mu_k^*\|^2 + \frac{d}{n_k}}} + o(1).
$$

As a result, if the total sample size is fixed, choosing large $n_k$ for small $\|\mu_k^*\|$ will make the average misclassification error small.

## B.3 Model performance analysis with Additional Metrics

In this section, we conduct additional analysis using commonly used metrics from vision-language models on the same dataset, and discuss the applicability of these methods to hallucination evaluation.

### B.3.1 Descriptions of Additional Metrics

**BLEU** BLEU (Bilingual Evaluation Understudy (Papineni et al., 2002)) is a metric used to evaluate the quality of machine-generated translations by comparing them to one or more reference translations. The BLEU score is based on the idea of precision in $n$-grams, which are contiguous sequences of $n$ words. It measures how well the generated translation matches the reference translations in terms of $n$-gram overlap.

**BertScore** BERTScore (Zhang et al., 2019) is a method for evaluating the quality of natural language generation or summarization systems. BERTScore measures the similarity between a reference text and a generated text by computing contextualized embeddings using BERT.

**ROUGE-L** ROUGE-L (Recall-Oriented Understudy for Gisting Evaluation - Longest Common Subsequence (Lin, 2004)) is an evaluation metric commonly used in natural language processing

and text summarization tasks. It is designed to measure the quality of a machine-generated summary by comparing it to one or more reference summaries.

**CLIP** CLIP (Contrastive Language-Image Pretraining (Radford et al., 2021)) score is a metric used to evaluate the performance of the vision-language model, which measures how well the model can correctly associate images with their corresponding captions or textual descriptions.

Besides, these four metrics, we further introduce METEOR, CIDER, and SPICE, which are detailed as follows:

**MENTOR** (Banerjee & Lavie, 2005): METEOR (Metric for Evaluation of Translation with Explicit ORdering) is a metric used to evaluate the performance of machine translation. It measures the extent to which a machine translation model can accurately associate the generated translation with its corresponding human reference translation.

**CIDER** (Vedantam et al., 2015): CIDER (Consensus-based Image Description Evaluation) is a metric used to assess the quality of image captioning models. It focuses on evaluating how well the generated captions align with human judgments.

**SPICE** (Anderson et al., 2016): SPICE (Semantic Propositional Image Caption Evaluation) is a metric used for evaluating the quality of image captions generated by machine models. Unlike traditional metrics that rely on n-gram matching, SPICE focuses on assessing the semantic similarity between the generated captions and human reference captions.

### B.3.2 RESULTS

In Table 10 and Table 11 (for METEOR, CIDER, and SPICE), we present the performance of different models and baselines on these metrics. Based on the experimental results, it is evident that LURE outperforms the other baselines in both text translation metrics and image-text matching metrics, with a notable improvement in the CLIP Score metric. This could be attributed to the higher sensitivity of the CLIP Score, as compared to text translation metrics like BLEU, in capturing object-level differences. These findings are consistent with the overall experimental results presented in Table 1, further confirming the effectiveness of LURE. However, we have also identified certain issues related to the BLEU metric for text translation. The differences between baselines were not very pronounced, possibly because such metrics tend to emphasize the evaluation of text style rather than object-level distinctions. These metrics may not be well-suited for assessing hallucinations and long-form descriptions when compared to CHAIR.

### B.4 ADDITIONAL RESULTS ON IMAGENET AND CC DATASETS

We conduct additional analyses to assess the performance of LURE on two newly introduced datasets: ImageNet (Deng et al., 2009) and CC (Conceptual Captions) (Changpinyo et al., 2021). Currently, the CHAIR metric can only be applied to the COCO dataset, which limits its usability beyond that dataset. To overcome this limitation, we manually annotate ImageNet and CC datasets to investigate object hallucination. Specifically, we randomly select 200 images from each dataset to be annotated. We evaluate the presence of hallucination in the generated captions through manual evaluation, using a scale where 0 indicated no hallucination and 1 indicated the presence of hallucination. The results presented in Table 12 demonstrate the performance improvements achieved by LURE across different datasets, thereby reinforcing our claims regarding LURE's effectiveness in reducing object hallucination in generated descriptions.

### B.5 ADDITIONAL RESULTS ON POPE AND MME

In addition to assessing the performance of our method, LURE, in mitigating hallucinatory objects in image captioning, we conduct additional experiments using LURE on other popular benchmark datasets, specifically MME (Fu et al., 2023) and POPE (Li et al., 2023d), as they are well-suited for evaluating hallucinations. For the POPE dataset, following the methodology of Li et al. (2023d), we conduct evaluations using LLaVa 13B. Since LURE is a post-hoc method, during testing, we incorporated the captions rectified by LURE as context in the prompts for reference to execute these tests. The final results are displayed in Table 13. For a fair comparison, we conducted additional experiments in Table 14 on these datasets by providing input in the form of the question along with

Table 10: Performance of different models and baselines on general metrics.

| Models | | BLEU-1 | BLEU-2 | BLEU-3 | BLEU-4 | BERTS | ROUGE-L | CLIPS |
|---|---|---|---|---|---|---|---|---|
| mPLUG-Owl | Original | 30.37 | 14.59 | 5.618 | 2.505 | **86.87** | 30.21 | 0.168 |
| | CoT | 25.04 | 11.48 | 4.229 | 1.954 | 86.61 | 29.86 | 0.189 |
| | Teacher | 29.91 | 14.22 | 5.519 | 2.431 | 86.76 | 31.15 | 0.192 |
| | Greedy-Decoding | 30.29 | 14.30 | 5.509 | 2.502 | 86.59 | 30.35 | 0.208 |
| | GPT-Ensemble | 29.74 | 13.91 | 5.121 | 2.367 | 85.94 | 28.90 | 0.159 |
| | GPT-Teacher | 28.19 | 14.13 | 6.181 | 3.128 | 86.65 | **30.87** | 0.215 |
| | **LURE (ours)** | **30.44** | **15.47** | **6.640** | **3.576** | 86.65 | 30.31 | **0.267** |
| LLaVa | Original | 30.88 | 15.46 | 6.984 | 3.586 | 86.96 | 31.53 | 0.242 |
| | CoT | 29.94 | 15.01 | 7.042 | 3.718 | 86.99 | 31.82 | 0.211 |
| | Teacher | 30.52 | 15.54 | 7.334 | 3.906 | 87.11 | 31.76 | **0.256** |
| | Greedy-Decoding | 31.76 | 17.21 | 8.491 | 4.223 | 87.01 | 32.50 | 0.249 |
| | GPT-Ensemble | 25.68 | 16.24 | 7.047 | 2.893 | 84.10 | 30.84 | 0.201 |
| | GPT-Teacher | 22.06 | 19.54 | 3.393 | 1.493 | 85.94 | 27.62 | 0.251 |
| | **LURE (ours)** | **35.94** | **21.81** | **11.33** | **6.804** | **87.39** | **32.59** | 0.238 |
| LLaMA-Adapter | Original | 29.95 | 15.36 | 7.324 | 3.875 | 86.83 | **31.77** | 0.179 |
| | CoT | 25.45 | 11.41 | 4.233 | 1.687 | 86.48 | 39.98 | 0.201 |
| | Teacher | 26.71 | 12.88 | 5.388 | 2.636 | 86.65 | 30.50 | 0.142 |
| | Greedy-Decoding | 30.66 | 14.63 | 6.920 | 2.309 | 86.90 | 31.69 | 0.211 |
| | GPT-Ensemble | 24.92 | 11.21 | 4.678 | 1.890 | 84.92 | 27.12 | 0.140 |
| | GPT-Teacher | 25.13 | 10.25 | 3.929 | 1.684 | 85.85 | 28.68 | 0.186 |
| | **LURE (ours)** | **30.94** | **15.81** | **7.334** | **3.804** | **86.96** | 31.60 | **0.223** |
| MiniGPT-4 | Original | 31.22 | 16.57 | 9.270 | 5.190 | 86.96 | 31.75 | 0.157 |
| | CoT | 33.68 | 20.57 | 10.72 | 6.430 | 86.09 | 32.39 | 0.177 |
| | Teacher | 32.69 | 19.87 | 9.870 | 5.350 | 86.06 | 30.72 | 0.142 |
| | Greedy-Decoding | 35.12 | 22.89 | 12.38 | 6.770 | 87.22 | 33.93 | 0.198 |
| | GPT-Ensemble | 29.65 | 19.22 | 9.878 | 5.330 | 85.77 | 29.83 | 0.140 |
| | GPT-Teacher | 33.37 | 20.28 | 11.52 | 5.770 | 87.01 | 31.89 | 0.182 |
| | **LURE (ours)** | **41.20** | **23.17** | **13.18** | **7.580** | **87.88** | **35.34** | **0.210** |
| MMGPT | Original | 27.27 | 12.66 | 5.680 | 2.290 | 79.79 | 29.03 | 0.177 |
| | CoT | 26.11 | 12.30 | 5.580 | 2.250 | 76.90 | 28.77 | 0.192 |
| | Teacher | 26.56 | 12.38 | 5.600 | 2.260 | 80.16 | 22.09 | 0.162 |
| | Greedy-Decoding | 30.15 | 15.11 | 6.320 | 3.573 | 86.62 | 31.77 | 0.188 |
| | GPT-Ensemble | 24.59 | 13.77 | 5.673 | 2.882 | 84.22 | 25.78 | 0.156 |
| | GPT-Teacher | 23.60 | 10.92 | 4.610 | 2.010 | 83.11 | 23.43 | 0.178 |
| | **LURE (ours)** | **32.71** | **16.24** | **7.407** | **3.830** | **87.01** | **32.31** | **0.201** |
| InstructBLIP | Original | 29.46 | 14.52 | 5.670 | 2.421 | 86.71 | 31.64 | 0.218 |
| | CoT | 24.04 | 12.61 | 4.086 | 1.837 | 85.50 | 28.07 | 0.229 |
| | Teacher | 25.61 | 12.22 | 4.321 | 1.963 | 85.93 | 29.89 | 0.294 |
| | Greedy-Decoding | 29.22 | 13.98 | 5.605 | 2.344 | 86.11 | 32.57 | 0.276 |
| | GPT-Ensemble | 26.32 | 13.11 | 5.101 | 2.396 | 85.04 | 30.77 | 0.198 |
| | GPT-Teacher | 24.91 | 11.92 | 4.652 | 2.097 | 85.81 | 29.49 | 0.205 |
| | **LURE (ours)** | **29.77** | **15.23** | **5.708** | **2.634** | **87.94** | **32.95** | **0.307** |

an original, uncorrected description of the image. This method is referred to as "Ori + Cap." For other methods, the input of "Original" consists of the original question and the corresponding image. For LURE, the input during inference comprises the original question, the image, and the description that has been rectified by LURE.

Furthermore, we evaluate three top-performing LVLMs with LURE on the Multimodal Model Evaluation (MME) benchmark (Fu et al., 2023). This benchmark comprises ten subtasks to evaluate models' perceptual capabilities and four subtasks for assessing cognitive abilities. To measure object hallucinations, we select a specific subset tailored for this purpose, similar to the POPE benchmark (Li et al., 2023d). This subset consists of a series of binary "Yes-or-No" questions. Following the evaluation settings used in the POPE benchmark, we employ metrics such as accuracy, recall, and F1 score to quantify the models' performance on this subset, with the results presented in Table 15.

Table 11: Performance on additional metrics – MENTOR, CIDER, SPICE.

| Models | | METEOR | CIDER | SPICE |
|---|---|---|---|---|
| mPLUG-Owl | Original | 28.7 | 0.53 | 17.5 |
| | **LURE** | **36.7** | **0.66** | **18.9** |
| LLaVa | Original | 37.7 | 0.61 | 22.6 |
| | **LURE** | **43.9** | **0.67** | **31.4** |
| LLaMA-Adapter | Original | 27.6 | 0.59 | 21.8 |
| | **LURE** | **33.4** | **0.63** | **29.2** |
| MiniGPT-4 | Original | 22.0 | 0.51 | 17.9 |
| | **LURE** | **25.6** | **0.55** | **26.4** |
| MMGPT | Original | 24.3 | 0.56 | 18.9 |
| | **LURE** | **26.8** | **0.61** | **20.1** |
| InstructBLIP | Original | 26.5 | 0.62 | 18.5 |
| | **LURE** | **30.3** | **0.72** | **19.6** |

Table 12: Results (human evaluation) on additional datasets - ImageNet and CC. We assessed hallucination in the generated captions through manual evaluation, employing a scale where 0 indicates the absence of hallucination, and 1 indicates its presence. The average hallucination ratio (%) is reported in this table.

| | | MiniGPT4 | LLaVA | LLaMA-Adapter | mPLUG-Owl |
|---|---|---|---|---|---|
| ImageNet | Original | 31.5 | 58.0 | 37.0 | 63.5 |
| | **LURE(ours)** | **22.5** | **24.0** | **28.5** | **32.0** |
| CC | Original | 23.5 | 36.0 | 41.0 | 52.5 |
| | **LURE(ours)** | **16.0** | **18.5** | **29.0** | **26.5** |

The results indicate a significant reduction in hallucination with the introduction of LURE in both the POPE and MME datasets. These findings not only highlight the effectiveness of LURE but also provide additional support for the conclusions drawn in our main paper.

Table 13: POPE results of LLaVa on MSCOCO, A-OKVQA, and GQA.

| Dataset | Model | POPE | Accuracy | Precision | Recall | F1 Score | Yes (%) |
|---|---|---|---|---|---|---|---|
| MSCOCO | LLaVa (Original) | Random | 54.43 | 52.32 | 99.80 | 68.65 | 95.37 |
| | | Popular | 52.43 | 51.25 | 99.80 | 67.72 | 97.37 |
| | | Adversarial | 50.77 | 50.39 | 99.87 | 66.98 | 99.10 |
| | LLaVa (**LURE**) | Random | 86.33 | 89.44 | 82.40 | 85.77 | 46.07 |
| | | Popular | 80.30 | 79.00 | 82.53 | 80.73 | 52.23 |
| | | Adversarial | 77.17 | 74.33 | 83.00 | 78.43 | 55.83 |
| A-OKVQA | LLaVa (Original) | Random | 50.16 | 50.08 | 99.53 | 66.64 | 99.37 |
| | | Popular | 50.03 | 50.02 | 99.67 | 66.61 | 99.63 |
| | | Adversarial | 50.13 | 50.07 | 99.67 | 66.65 | 99.53 |
| | LLaVa (**LURE**) | Random | 83.70 | 84.32 | 82.80 | 83.55 | 49.10 |
| | | Popular | 78.00 | 75.86 | 82.13 | 78.87 | 54.13 |
| | | Adversarial | 69.93 | 65.72 | 83.33 | 73.49 | 63.40 |
| GQA | LLaVa (Original) | Random | 50.17 | 50.08 | 99.20 | 66.56 | 99.03 |
| | | Popular | 50.03 | 50.02 | 99.47 | 66.56 | 99.43 |
| | | Adversarial | 49.77 | 49.88 | 99.20 | 66.38 | 99.43 |
| | LLaVa (**LURE**) | Random | 83.32 | 84.22 | 82.47 | 83.25 | 49.15 |
| | | Popular | 80.85 | 80.09 | 82.47 | 81.20 | 51.62 |
| | | Adversarial | 78.74 | 76.67 | 82.77 | 79.58 | 54.03 |

Table 14: POPE results of LLaVa on A-OKVQA, and GQA. "LLaVa (Ori + Cap)" indicates that during testing, we provided the input as the question along with LLaVa's original, uncorrected description of the image.

| Dataset | Model | POPE | Accuracy | Precision | Recall | F1 Score | Yes (%) |
|---|---|---|---|---|---|---|---|
| A-OKVQA | LLaVa (Original) | Random | 50.16 | 50.08 | 99.53 | 66.64 | 99.37 |
| | | Popular | 50.03 | 50.02 | 99.67 | 66.61 | 99.63 |
| | | Adversarial | 50.13 | 50.07 | 99.67 | 66.65 | 99.53 |
| | LLaVa (Ori + Cap) | Random | 74.93 | 75.20 | 91.17 | 74.17 | 61.28 |
| | | Popular | 70.01 | 68.94 | 90.32 | 73.57 | 63.23 |
| | | Adversarial | 65.13 | 62.40 | 92.17 | 68.87 | 72.38 |
| | LLaVa (**LURE**) | Random | 83.70 | 84.32 | 82.80 | 83.55 | 49.10 |
| | | Popular | 78.00 | 75.86 | 82.13 | 78.87 | 54.13 |
| | | Adversarial | 69.93 | 65.72 | 83.33 | 73.49 | 63.40 |
| GQA | LLaVa (Original) | Random | 50.17 | 50.08 | 99.20 | 66.56 | 99.03 |
| | | Popular | 50.03 | 50.02 | 99.47 | 66.56 | 99.43 |
| | | Adversarial | 49.77 | 49.88 | 99.20 | 66.38 | 99.43 |
| | LLaVa (Ori + Cap) | Random | 75.23 | 73.66 | 90.37 | 74.73 | 60.38 |
| | | Popular | 75.12 | 73.24 | 90.32 | 74.47 | 60.59 |
| | | Adversarial | 67.63 | 63.40 | 83.17 | 70.18 | 65.19 |
| | LLaVa (**LURE**) | Random | 83.32 | 84.22 | 82.47 | 83.25 | 49.15 |
| | | Popular | 80.85 | 80.09 | 82.47 | 81.20 | 51.62 |
| | | Adversarial | 78.74 | 76.67 | 82.77 | 79.58 | 54.03 |

Table 15: Performance comparison before and after applying LURE on MME. Since we found that TN (True Negatives) and FN (False Negatives) are both zero in the MME dataset, the values of accuracy and recall are the same.

| Models | | Accuracy | Recall | F1 Score |
|---|---|---|---|---|
| LLaVa | Original | 90.0 | 90.0 | 94.7 |
| | **LURE** | **93.3** | **93.3** | **96.6** |
| MiniGPT-4 | Original | 93.8 | 93.8 | 96.8 |
| | **LURE** | **96.7** | **96.7** | **98.3** |
| mPLUG-Owl | Original | 86.7 | 86.7 | 92.6 |
| | **LURE** | **93.5** | **93.5** | **96.7** |

## C  ADDITIONAL ANALYSIS OF LURE

### C.1  ADDITIONAL ANALYSIS ABOUT THE HULLUCINATION FACTORS

#### C.1.1  ADDITIONAL ANALYSIS OF OBJECT POSITIONS AND HALLUCINATIONS

To gain a deeper understanding of the impact of object position on hallucinations, we extend our analysis beyond the existing evaluation presented in Figure 1c. This extended analysis encompasses the following evaluations:

- **Evaluation with More Examples.** In the first phase of our evaluation, we re-assess the distribution of hallucinatory objects concerning their positions using a larger dataset comprising 5,000 examples from the COCO dataset. The results are detailed in Figure 5a.

- **Evaluation on Short Descriptions.** Similarly, in the second phase, we evaluate the distribution of hallucinatory objects concerning their positions within short descriptions generated by models such as OFA, BLIP2, etc., using the same 5,000 data points as in the first evaluation. These findings are illustrated in Figure 5b in our updated paper.

- **Evaluation on Other Datasets.** In the third phase, we explore the relationship between the distribution of hallucinatory objects and their positions in ImageNet and CC dataset Deng et al. (2009);

Sharma et al. (2018). For this evaluation, descriptions are manually annotated to identify hallucinated objects, and the results are reported in Figure 5c.

Across all evaluations, our findings consistently indicate that high-density areas of hallucinatory objects predominantly appear towards the end of the sequence, regardless of the length of the descriptions. This further reinforces our original conclusions. Furthermore, it is worth noting that generating shorter descriptions does not yield lower position hallucination. Therefore, simply generating multiple short descriptions and combining them may not necessarily lead to higher-quality descriptions.

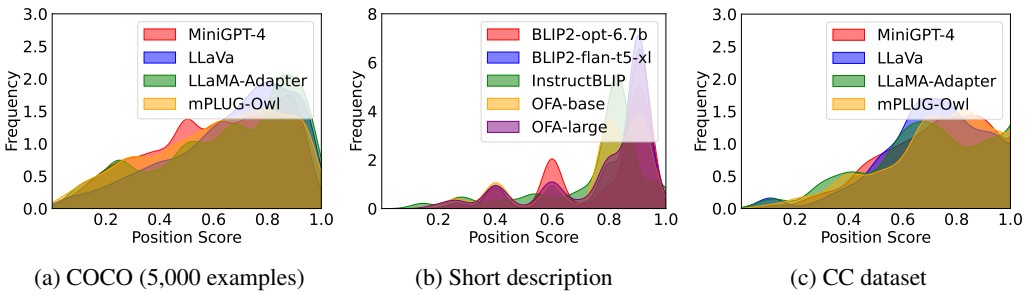

(a) COCO (5,000 examples)    (b) Short description    (c) CC dataset

Figure 5: Additional analysis of the relationship between object position and hallucination.

### C.1.2 CAN LURE REDUCE OBJECT HALLUCINATIONS CAUSED BY CO-OCCURRENCE, UNCERTAINTY AND OBJECT POSITION?

To validate that our method reduces co-occurrence, uncertainty, and object positional bias that affect object hallucination, we further verify by evaluating the proportion of hallucinatory objects in high uncertainty, high co-occurrence, and sentence-ending positions. We compared the changes in various proportions of descriptions using MiniGPT-4 and LURE on the COCO 2014 test dataset. Here, we first describe how we calculate the object ratio under different factors:

**Ratio of Co-occurrence-Based Hallucinatory Objects.** Similiar to uncertainty hallucination ratio, we obtain the $C_{ratio}$ by calculating ratio of the number of hallucination objects with high co-occurence score and the total number of objects with high co-occurence score:

$$C_{ratio} = \frac{\sum_{s=1}^{M_h} \mathbb{1}[\text{CoScore}_s \geq \text{CoScore}_{mean}]}{\sum_{m=1}^{M} \mathbb{1}[\text{CoScore}_m \geq \text{CoScore}_{mean}]}, \tag{5}$$

where $M_h$ is the number of hallucinatory descriptions, $M$ represents the number of total descriptions, and $\text{CoScore}_{mean} = \frac{1}{M} \sum_{m=1}^{M} \text{CoScore}_m$.

**Ratio of Uncertainty-Based Hallucinatory Objects.** We obtain the $U_{ratio}$ by calculating ratio of the number of hallucination objects with high uncertainty and the total number of objects with high uncertainty:

$$U_{ratio} = \frac{\sum_{s=1}^{M} \sum_{i=1}^{n_h} \mathbb{1}[\text{UnScore}_{s,i} \geq \text{UnScore}_{mean}]}{\sum_{m=1}^{M} \sum_{j=1}^{n_h+n_r} \mathbb{1}[\text{UnScore}_{m,j} \geq \text{UnScore}_{mean}]}, \tag{6}$$

where

$\text{UnScore}_{mean} = \frac{1}{M(n_h+n_r)} \sum_{m=1}^{M} \sum_{j=1}^{n_h+n_r} \text{UnScore}_{m,j}$.

**Ratio of Hallucinatory Objects in Later Part of the Sentence.** For the ratio of hallucinatory objects in later part of the sentence., we calculate the $S_{ratio}$ by calculating ratio of the number of hallucination objects in later part of the sentence and the total number of objects in later part of the sentence:

$$S_{ratio} = \frac{\sum_{s=1}^{M} \sum_{i=1}^{n_h} \mathbb{1}[\text{PoScore}_{s,i} \geq \eta]}{\sum_{m=1}^{M} \sum_{i=1}^{n_h+n_r} \mathbb{1}[\text{PoScore}_{m,i} \geq \eta]}, \tag{7}$$

Table 16: Uncertainty-based hallucination object ratio, co-occurrence-based hallucination object ratio, and sentence-ending hallucination object ratio analysis on several models.

| Models | | Co-occurrence $C_{Ratio}$ | Uncertainty $U_{Ratio}$ | Position $S_{Ratio}$ |
|---|---|---|---|---|
| MiniGPT-4 | Original | 0.106 | 0.221 | 0.227 |
| | **LURE (ours)** | **0.071** | **0.145** | **0.139** |
| LLaVa | Original | 0.243 | 0.103 | 0.331 |
| | **LURE (ours)** | **0.142** | **0.086** | **0.139** |
| LLaMA-Adapter | Original | 0.295 | 0.178 | 0.442 |
| | **LURE (ours)** | **0.176** | **0.102** | **0.272** |
| mPLUG-Owl | Original | 0.128 | 0.229 | 0.259 |
| | **LURE (ours)** | **0.106** | **0.127** | **0.151** |
| MMGPT | Original | 0.110 | 0.157 | 0.418 |
| | **LURE (ours)** | **0.089** | **0.114** | **0.154** |
| InstructBLIP | Original | 0.213 | 0.147 | 0.389 |
| | **LURE (ours)** | **0.123** | **0.090** | **0.156** |

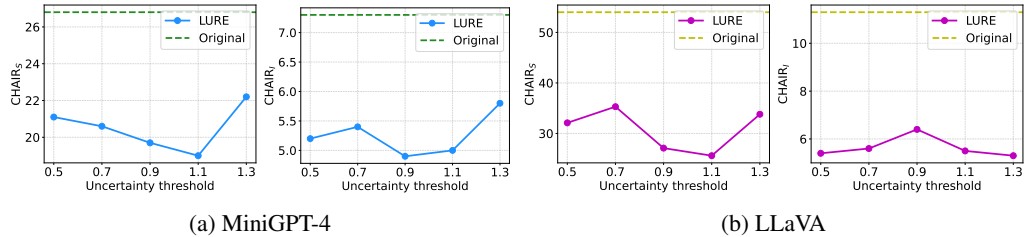

(a) MiniGPT-4              (b) LLaVA

Figure 6: Sensitivity analysis of uncertainty threshold using MiniGPT-4 and LLaVA as revisor backbone.

where $\eta$ is the position threshold.

**Results.** Based on the data presented in Table 16, it is evident that all three categories of ratios in the descriptions of LURE reduce when compared to the ratios of the original descriptions. This observation indicates that the elements of uncertainty, co-occurrence, and object position have contributed less to hallucinations in LURE.

### C.1.3 PARAMETER SENSITIVITY ANALYSIS ON UNCERTAINTY

To further illustrate the robustness of our model, we conducted a parameter sensitivity analysis on the threshold of uncertainty. The uncertainty threshold $\theta$ determines the proportion of replacing "object" with [IDK]. From the Figure 6, we can observe that our model is robust within a certain range of uncertainty threshold.

### C.2 DOES APPLYING LURE AFFECT THE USEFULNESS?

We conduct additional analyses to examine the impact of LURE on the diversity and completeness of descriptions generated by various models before and after applying LURE. Our primary focus is on several key aspects: changes in description length, reduction in the proportion of correctly identified objects, and reduction in hallucinatory objects after applying LURE. The detailed results for six LVLMs using the same dataset as the main paper are presented in Table 17.

Our findings reveal that the incorporation of LURE leads to a significant reduction in hallucinatory objects, averaging around 56%, while only slightly affecting the presence of correctly identified objects, with an average decrease of approximately 1.6%. This noteworthy outcome can be attributed to the fact that LURE doesn't merely eliminate potentially hallucinatory objects; it actively encourages the model to reconsider and either remove or replace such objects. This approach significantly enhances model performance and reduces hallucination. Furthermore, the positive effect of LURE is evident in the average length of the generated descriptions. Applying LURE results in only minor

changes to the description length, indicating its effectiveness in preserving the utility and diversity of the generated responses. In summary, the use of LURE achieves a balance between the correctness and usefulness of responses in LVLMs.

Table 17: Analysis of correctness and usefulness before and after applying LURE.

|  | MiniGPT-4 | LLaVa | MMGPT | LLaMA-Adapter | mPLUG-Owl | InstructBLIP |
|---|---|---|---|---|---|---|
| Correct Decrease Reduction of correct objects (%) | 0.680 | 2.150 | 1.420 | 1.610 | 1.130 | 2.720 |
| Reduction of hallucinated object (%) | 41.13 | 56.51 | 69.36 | 52.19 | 61.88 | 54.96 |
| Average description length (before) | 67.08 | 102.8 | 63.18 | 94.27 | 110.1 | 95.63 |
| Average description length (after) | 56.63 | 96.39 | 57.24 | 93.44 | 99.15 | 92.27 |

### C.3 CAN LURE REDUCE OBJECT HALLUCINATION IN SHORT DESCRIPTIONS?

To further explore the effectiveness of LURE with concise captions, we conduct additional experiments, the results of which are presented in Table 18. The concise descriptions are generated using the prompt "Generate a short caption of the image." Our findings indicate that LURE remains effective in reducing object hallucinations even with shorter captions, thus reinforcing its capability in mitigating such issues.

Table 18: Performance of LURE on short descriptions generated by the four best-performing LVLMs.

|  | MiniGPT-4 | | LLaVa | | LLaMA-Adapter | | mPLUG-Owl | |
|---|---|---|---|---|---|---|---|---|
|  | $C_S\downarrow$ | $C_I\downarrow$ | $C_S\downarrow$ | $C_I\downarrow$ | $C_S\downarrow$ | $C_I\downarrow$ | $C_S\downarrow$ | $C_I\downarrow$ |
| Original | 18.4 | 7.6 | 33.3 | 10.5 | 23.6 | 8.4 | 14.6 | 7.1 |
| **LURE (ours)** | **10.6** | **3.1** | **21.2** | **5.3** | **20.2** | **5.4** | **13.1** | **3.7** |

## D ADDITIONAL CASE STUDIES

### D.1 CASES OF UNCERTAINTY

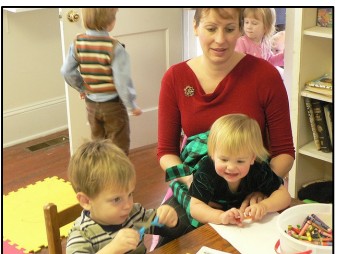

This image depicts a group of people sitting around a table. The people are wearing different clothes. There is a window in the background, and the room appears to be well-lit. The walls of the room are painted white and there are two doorways that lead to other rooms.

**The probability of the vocabulary within the red box**

book: 0.17, window: 0.15, child: 0.13, wooden: 0.09, small: 0.07 ......

Figure 7: Case of uncertainty in the MiniGPT-4.

We provide an example using MiniGPT-4 to illustrate the uncertainty present in LVLMs during the decoding process. In the example, we display the word probabilities in the vocabulary at the location of hallucinatory words (sorted in descending order of probability). As shown in Figure 7, we have displayed the decoded tokens and their probabilities at the point where the hallucinatory word "window" occurs. We can observe that the probability of the hallucinatory word "window" is comparable to that of "book". The uncertainty in the model's decoding path is highly influenced by the text generated earlier, leading to the incorrect selection of the word "window" when generating this token.

### D.2 CASES OF OUR TRAINING DATASET

Here, we present some cases of training data constructed using GPT-3.5, as shown in Table 19. "Original caption" represents the original standard description, while the "Hallucination caption"

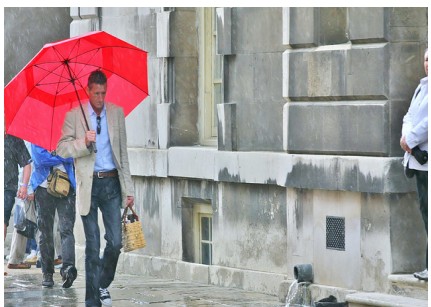

**Original Caption:** The image shows a man walking down a rainy sidewalk while holding a bright red umbrella to stay dry. The man walks next to a building as rain pours down, making the umbrella a necessary accessory. In addition to the man with the red umbrella, there are several other people in the scene, some of which are carrying handbags despite the wet conditions. Towards the edge of the image, a person holding a small umbrella can be seen, indicating that others are also trying to shield themselves from the rain. The busy street with multiple passersby creates an atmosphere of a bustling city adapting to the rainy weather.

**Hallucination Caption:** The image shows a man walking down a rainy sidewalk while holding a bright red umbrella to stay dry. Cars driving by with their headlights on contribute to the bustling city atmosphere. The man walks next to a building as rain pours down, making the umbrella a necessary accessory. In addition to the man with the red umbrella, there are several other people in the scene, some of which are carrying handbags despite the wet conditions. Towards the edge of the image, a person holding a small umbrella can be seen, indicating that others are also trying to shield themselves from the rain. The busy street with multiple passersby creates an atmosphere of a bustling city adapting to the rainy weather.

(a) Case 1.

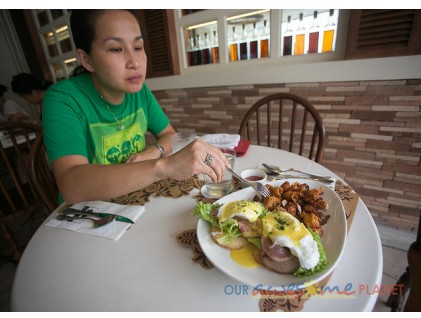

**Original Caption:** The image depicts a woman in a green shirt sitting at a dining table eating a meal. She is using a fork to pick at the food on her plate, which is positioned right in front of her. The table is set with a variety of items like a sandwich, a bowl, and multiple utensils such as knives and spoons. There are also several cups placed on the table. Although there are other chairs around the table, the woman appears to be dining alone, adding a sense of solitude to the scene. Other empty chairs can be seen in various positions around the dining table. Additionally, there are a few other individuals in the background, but they don't seem to be engaging with the woman or her meal.

**Hallucination Caption:** The image depicts a woman in a green shirt sitting at a dining table eating a meal. She is using a fork to pick at the food on her plate, which is positioned right in front of her. The table is set with a variety of items like a sandwich, fries, a bowl, and multiple utensils such as knives and spoons. There are also several cups placed on the table. A napkin and water glass are neatly placed beside her plate. Although there are other chairs around the table, the woman appears to be dining alone, adding a sense of solitude to the scene. Other empty chairs can be seen in various positions around the dining table. Additionally, there are a few other individuals in the background, but they don't seem to be engaging with the woman or her meal. The salt and pepper shakers are placed at the center of the table, within easy reach of the woman.

(b) Case 2.

Table 19: Cases of generating hallucinatory descriptions.

column represents the hallucinated description constructed by GPT-3.5. The red portions in the hallucination captions indicate the hallucinations added by GPT-3.5 based on co-occurring object lists and uncertain object lists.

### D.3 CASES OF REWRITING CAPTIONS

In this section, we present several examples of rectified descriptions to demonstrate the capabilities of LURE in reducing hallucination. From 9 we can find that our model demonstrates a high level of proficiency in removing or substituting hallucinatory objects.

### D.4 ADDITIONAL CASE COMPARISON BETWEEN LURE AND BASELINES

We carefully selected several baselines that demonstrated promising performance based on our experimental results and conducted a thorough comparison with our proposed method. The detailed results of this comparison can be found in Figure 10. Upon comparing the descriptions generated by Revisior with those from the other methods, it becomes evident that Revisior surpasses the others in terms of accuracy and level of detail in describing the image.

The description produced by Revisior effectively captures the key elements of the image, such as the presence of a man wearing a white shirt walking on the tennis court while holding a tennis racket, as well as the presence of other individuals in the scene. On the contrary, the other methods fall short in various aspects. The "Original" method's description includes numerous hallucinated objects like the "net" and "cap." Although the "CoT" method's description has fewer hallucinated objects, it is observed that errors in the step-by-step reasoning process, such as incorrectly stating the presence of two tennis players, lead to corresponding errors in subsequent descriptions.

While the "Teacher" method's description is somewhat accurate, it still struggles to eliminate hallucinated objects effectively. Although GPT demonstrates strong textual comprehension abilities, it can still make mistakes when rewriting descriptions due to the absence of visual patterns, resulting in the omission of hallucinated objects and introducing errors.

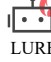

<Input image>

Describe this image 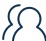

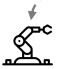

LURE

The image features a woman playing tennis on a clay court. She is in the middle of a swing, holding a tennis racket and preparing to hit the ball. The woman is wearing a pink dress and white shoes, which complement her athletic attire. There are several other people in the scene, likely spectators or fellow players, positioned around the court. Some of them are standing closer to the woman, while others are further away, observing the game. The main focus of the image remains the woman tennis player in action.

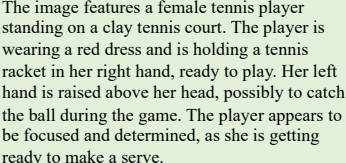

Under revision!!!

The image features a female tennis player standing on a clay tennis court. The player is wearing a red dress and is holding a tennis racket in her right hand, ready to play. Her left hand is raised above her head, possibly to catch the ball during the game. The player appears to be focused and determined, as she is getting ready to make a serve.

<Input image>

Describe this image 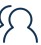

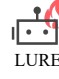

LURE

The image captures a man skillfully riding a wave on a surfboard in the ocean. He is crouching down on his board, maintaining balance and control as he navigates the wave. The surfer is positioned towards the center of the image, with the wave surrounding him. In the background, there are several other people in the water, possibly observing or waiting for their turn to surf. Some of them are closer to the shore, while others are further out in the ocean. The scene conveys the excitement and challenge of surfing, as well as the camaraderie among those who share this passion.

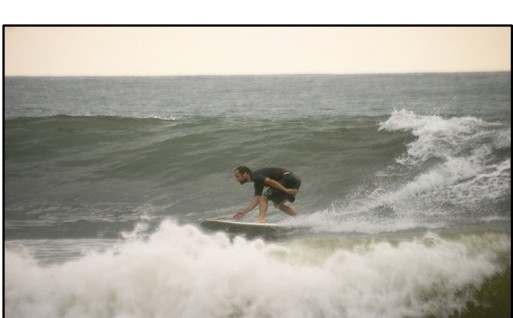

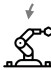

Under revision!!!

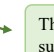

The image shows a person standing on a surfboard in the ocean, riding a large wave. The person is wearing a black shirt and black pants and is positioned on top of the surfboard as he rides the wave. The wave is large and has a lot of water coming off of it. The person appears to be enjoying the ride.

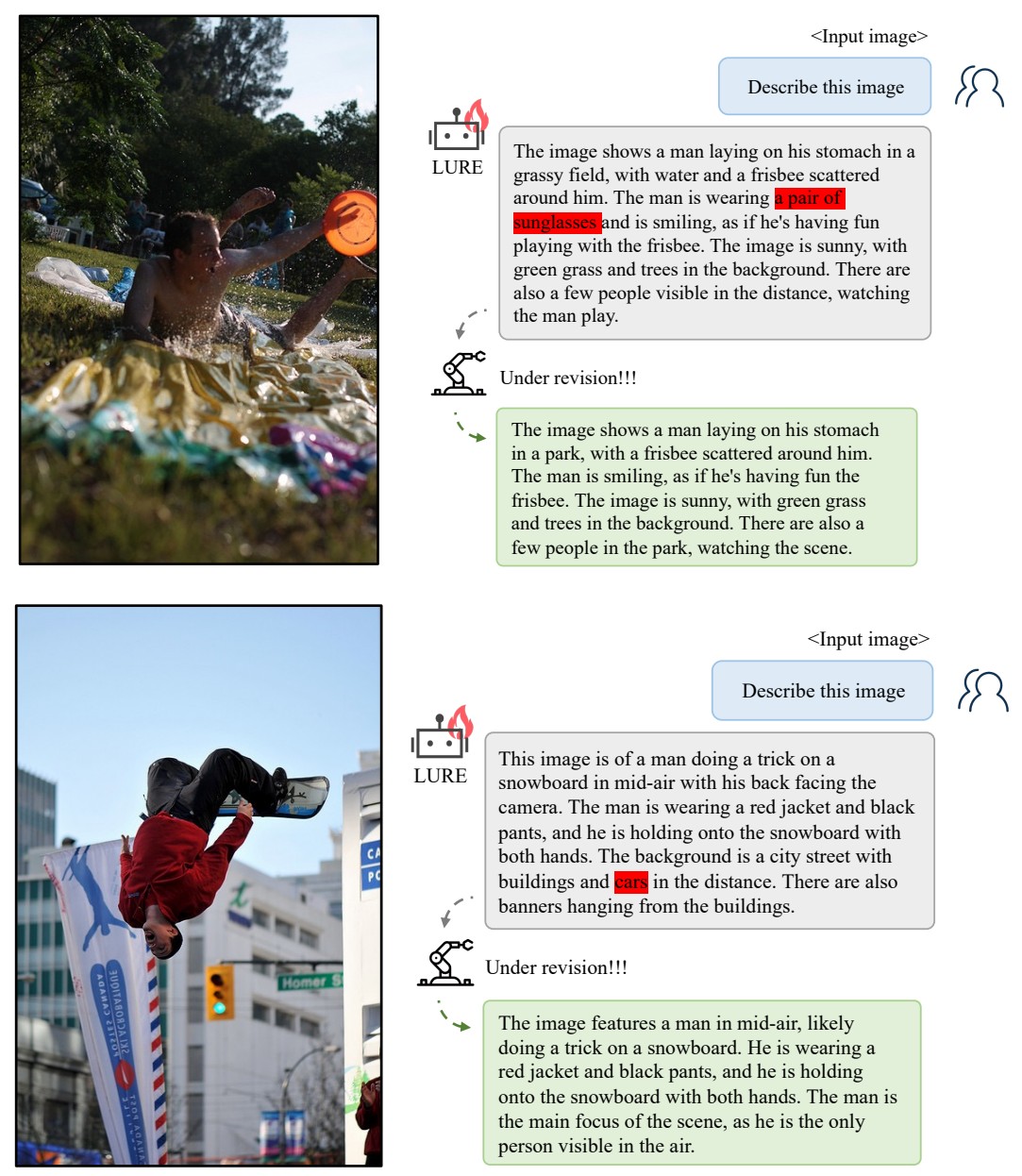

Figure 9: Additional cases of rectified descriptions

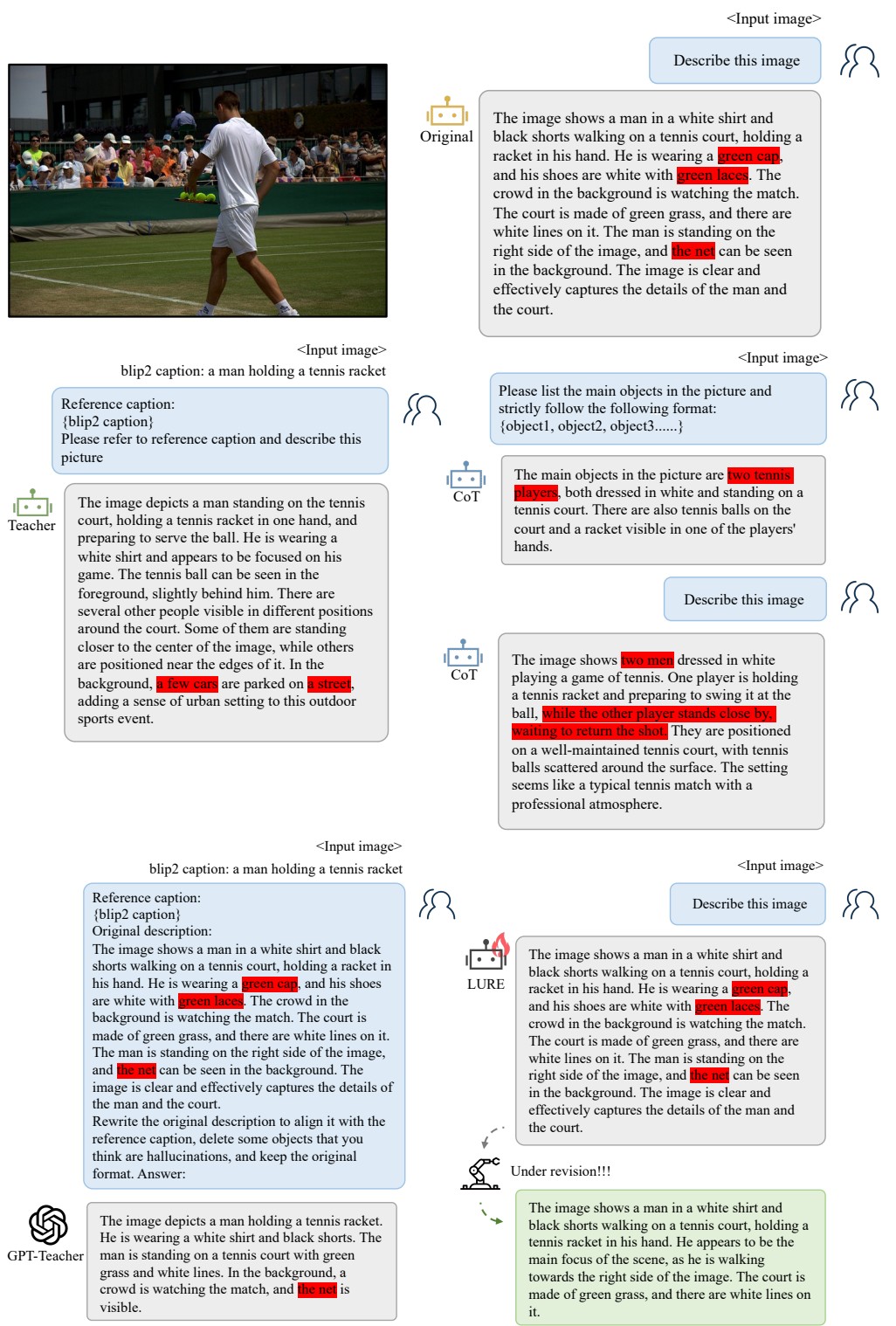

Figure 10: Case study of several strong baselines, including detailed dialogue flow of the real inquiry process for each baseline.

