# OpenReview forum: "Analyzing and Mitigating Object Hallucination in Large Vision-Language Models"
_ICLR.cc/2024/Conference — ICLR 2024 poster_

### Official Review · Reviewer_ZMPq · 2023-10-20

**Soundness:** 3 good
**Presentation:** 2 fair
**Contribution:** 3 good
**Rating:** 6
**Confidence:** 3

**Summary:**

This paper studies the problem of object hallucination in large vision and language models. They first analyze the patterns and relations between object hallucinations and three concepts. Then, they provide theoretical analysis and explanation for the observations. Based on the analysis, they create a dataset for training a caption revising model to mitigate the hallucination in captions. The experiment results show that the caption after revising has fewer object hallucinations than the original caption generated by LVLMs and outperforms several baselines.

**Strengths:**

1. The paper conducts an early study on the caption object hallucination problem of LVLMs and provides some analysis, observations, and theoretical analysis on object hallucination. The findings are meaningful to future research.
2. Based on the analysis, the paper proposes a simple and effective method to mitigate caption object hallucination by training a caption revising model.
3. The paper tests the proposed method on multiple LVLMs and different metrics and compares it with different baselines. The results validate the effectiveness of the proposed method.

**Weaknesses:**

1. The study and the proposed method are limited to caption hallucination problems, and seem not generalized to other settings like VQA.
2. Both the training data of the hallucination revisor and the testing data are from COCO datasets. Whether the proposed method can be generalized to new datasets with object labels needs to be validated.

**Questions:**

The reviewer has some questions on the theoretical analysis part:
1. In the analysis of **Co-occurrence**, can the authors please explain what is the meaning and why $fˆ_{2} = ⟨ϕ_{1}(s<i, x), βˆ_{1}⟩+⟨ϕ_{2}(s<i, x), βˆ_{2}⟩$? (which means that $fˆ_{2} = fˆ_{1}+⟨ϕ_{2}(s<i, x), βˆ_{2}⟩$)
2. The reviewer understands how the proposed methods related to the three observations on the object hallucinations. However, the reviewer doesn't see a clear connection between the theoretical analysis and the proposed methods. Can the authors explain this point?

---

> ### Author Response · Authors · 2023-11-19
> **Response to Reviewer ZMPq (1/3)**
>
> Thank you for your valuable feedback to help us improve our paper. We have revised our paper based on your feedback. We detail our response below and please kindly let us know if our response addresses your concerns.
>
> > **Q1**: The study and the proposed method are limited to caption hallucination problems, and seem not generalized to other settings like VQA.
>
> **A1**:  To strengthen the generalization of LURE, we conduct additional experiments on other popular benchmarks containing VQA (Visual Question Answering) questions, specifically on MME [Fu et al., 2023] and POPE [Li et al., 2023]. Since LURE is a post-hoc rectification method, during testing, we incorporated the captions rectified by LURE as context in the prompts for reference to run these VQA evaluations. The results in Tables R1 and R2 consistently demonstrate that incorporating LURE significantly reduces hallucination in VQA problems on MME and POPE (see detailed analysis in Appendix B.5). In the future, we aim to extend LURE to more complex VQA settings.
>
> **Table R1**:  POPE results of LLaVa on MSCOCO, A-OKVQA, and GQA.
>
> | Dataset    | Model            | Evaluation Setting | Accuracy | Precision | Recall | F1 Score | Yes (%) |
> |------------|------------------|--------------------|----------|-----------|--------|----------|---------|
> | A-OKVQA    |                  | Random             | 50.16    | 50.08     | 99.53  | 66.64    | 99.37   |
> |            | LLaVa (Original) | Popular            | 50.03    | 50.02     | 99.67  | 66.61    | 99.63   |
> |            |                  | Adversarial        | 50.13    | 50.07     | 99.67  | 66.65    | 99.53   |
> |            |                  | Random             | **83.70**    | **84.32**     | 82.80  | **83.55**    | 49.10   |
> |            | **LLaVa (LURE)**     | Popular            | **78.00**    | **75.86**     | 82.13  | **78.87**    | 54.13   |
> |            |                  | Adversarial        | **69.93**    | **65.72**     | 83.33  | **73.49**    | 63.40   |
> | GQA        |                  | Random             | 50.17    | 50.08     | 99.20  | 66.56    | 99.03   |
> |            | LLaVa (Original) | Popular            | 50.03    | 50.02     | 99.47  | 66.56    | 99.43   |
> |            |                  | Adversarial        | 49.77    | 49.88     | 99.20  | 66.38    | 99.43   |
> |            |                  | Random             | **83.32**    | **84.22**     | 82.47  | **83.25**    | 49.15   |
> |            | **LLaVa (LURE)**     | Popular            | **80.85**    | **80.09**     | 82.47  | **81.20**    | 51.62   |
> |            |                  | Adversarial        | **78.74**    | **76.67**     | 82.77  | **79.58**    | 54.03   |
>
>
>
> **Table R2**: Hallucinatory performance of MME before and after the correction by LURE. Since we found that TN (True Negatives) and FN (False Negatives) are both zero in the MME dataset, the values of accuracy and recall are the same.
> | Models         | Version | Accuracy | Recall | F1 Score |
> |----------------|---------|----------|--------|----------|
> | LLaVa      | Original| 90.0     | 90.0   | 94.7     |
> |                | **LURE (ours)**    | **93.3** | **93.3** | **96.6** |
> | MiniGPT-4  | Original| 93.8     | 93.8   | 96.8     |
> |                | **LURE (ours)**    | **96.7** | **96.7** | **98.3** |
> | Mplug-Owl  | Original| 86.7     | 86.7   | 92.6     |
> |                | **LURE (ours)**    | **93.5** | **93.5** | **96.7** |

---

> ### Author Response · Authors · 2023-11-19
> **Response to Reviewer ZMPq (2/3)**
>
> > **Q2**:  Both the training data of the hallucination revisor and the testing data are from COCO datasets. Whether the proposed method can be generalized to new datasets with object labels needs to be validated.
>
>
> **A2**:  We conduct additional analyses to assess the performance of LURE on two newly introduced datasets: ImageNet [Deng et al., 2009] and CC (Conceptual Captions) [Piyush et al.,  2018]. Currently, the CHAIR metric can only be applied to the COCO dataset, which limits its usability beyond that dataset. To overcome this limitation, we manually annotate ImageNet and CC datasets to investigate object hallucination. Specifically, we randomly select 200 images from each dataset to be annotated. We evaluate the presence of hallucination in the generated captions through manual evaluation, using a scale where 0 indicated no hallucination and 1 indicated the presence of hallucination. The results presented in Table R3 demonstrate the performance improvements achieved by LURE across different datasets, thereby reinforcing our claims regarding LURE's effectiveness in reducing object hallucination in generated descriptions. These findings have been incorporated into our updated paper and can be found in Appendix B.5.
>
> **Table R3**: Results (human evaluation) on additional datasets - ImageNet and CC. We assessed hallucination in the generated captions through manual evaluation, employing a scale where 0 indicates the absence of hallucination, and 1 indicates its presence. The average hallucination ratio (%) is reported in this table.
>
> |             |             | MiniGPT-4 | LLaVA | LLaMA-Adapter | mPLUG-Owl |
> |-------------|-------------|----------|-------|---------------|-----------|
> |             | Original    | 31.5     | 58.0  | 37.0          | 63.5      |
> | ImageNet    | **LURE (ours)**   | **22.5**| **24.0**| **28.5**     | **32.0**  |
> |             | Original    | 23.5     | 36.0  | 41.0          | 52.5      |
> | CC          | **LURE (ours)***   | **16.0**| **18.5**| **29.0**     | **26.5**  |
>
> ---
>
> >**Q3** In the analysis of Co-occurrence, can the authors please explain what is the meaning and why $\hat f_2=\langle  \phi_1(s_{<i},x),\hat\beta_1\rangle+\langle\phi_2(s_{<i},x),\hat\beta_2\rangle$?
>
> **A3**: Thank you for your comment. We would like to note that $\langle\phi_1(s_{<i},x),\hat\beta_1\rangle+\langle\phi_2(s_{<i},x),\hat\beta_2\rangle=\langle (\phi_1(s_{<i},x), \phi_2(s_{<i},x)), (\hat\beta_1, \hat\beta_2)\rangle$. We use the enriched feature vector $(\phi_1(s_{<i},x), \phi_2(s_{<i},x))$ for the second object prediction to model the sequential prediction manner in auto-regressive inferences. The $\hat\beta_1$ in $\hat f_1$ and $\hat f_2$ are not necessarily equal. They are equal in our proof by coincidence as the solutions (to the linear discriminant analysis rule) of $\hat\beta_1$’s for $\hat f_1$ and $\hat f_2$ happen to be equal under our model assumptions.
>
> ---
>
> >**Q4** The reviewer understands how the proposed methods related to the three observations on the object hallucinations. However, the reviewer doesn't see a clear connection between the theoretical analysis and the proposed methods. Can the authors explain this point?
>
> **A4**: LURE can be conceptualized as an augmentation of the original LVLM, achieved by introducing a revisor as the last layer within the LVLM framework. Here, the input image is also used as an input for this revisor layer. Subsequently, we engage in a fine-tuning process specifically targeting the revisor component. During the fine-tuning phase, our objective is to address the issue of hallucinations that may arise within the LVLM. This is achieved by fine-tuning the revisor using carefully curated training data that is aware of the co-occurrence and uncertainty issue. In our theoretical analysis, we demonstrate that incorporating such data can significantly enhance prediction accuracy. For example, consider the “reconsidering uncertain objects” step of LURE. This step involves considering more data points with large uncertainty in the curated dataset. According to Theorem 2.2, this adjustment is proven to have a positive impact on prediction accuracy. We have elaborated on these connections between the theoretical analysis and the proposed method further in Section 2.4 of the revised paper.

---

> > ### Author Response · Authors · 2023-11-19
> > **Response to Reviewer ZMPq (3/3)**
> >
> > **Reference**
> >
> > [Fu et al., 2023] Fu, Chaoyou, et al. "MME: A Comprehensive Evaluation Benchmark for Multimodal Large Language Models." arXiv preprint arXiv:2306.13394 (2023).
> >
> > [Li et al., 2023] Li, Yifan, et al. "Evaluating object hallucination in large vision-language models." arXiv preprint arXiv:2305.10355 (2023).
> >
> > [Deng et al., 2009] Imagenet: A large-scale hierarchical image database. In 2009 IEEE Conference on Computer Vision and Pattern Recognition (pp. 248–255)
> >
> > [Piyush  et al., 2018] Conceptual Captions: A Cleaned, Hypernymed, Image Alt-text Dataset For Automatic Image Captioning. In Proceedings of the 56th Annual Meeting of the Association for Computational Linguistics (Volume 1: Long Papers), pages 2556–2565, Melbourne, Australia. Association for Computational Linguistics​

---

> > > ### Author Response · Authors · 2023-11-21
> > > **We would like to hear back from reviewer ZMPq**
> > >
> > > Dear reviewer ZMPq,
> > >
> > > We would like to follow up to see if the response addresses your concerns or if you have any further questions. We would really appreciate the opportunity to discuss this further if our response has not already addressed your concerns. Thank you again!

---

> ### Comment · Reviewer_ZMPq · 2023-11-22
>
> Thanks a lot to the authors for the detailed response and the extra experiments. In the results in Table R1, the reviewer is not sure whether the following understanding is correct: The input to the LLaVA baseline is image embeddings, and the input to the LLaVa (LURE) includes the caption corrected by LURE and the image.

---

> ### Author Response · Authors · 2023-11-22
> **Additional Response to Reviewer ZMPq**
>
> Dear Reviewer ZMPq,
>
> Thank you very much for your response.
>
> In the experiments presented in Table R1, during the inference process, for **LLaVa (Original)**, the input to LLaVa 13B consists of the original question from POPE and the corresponding image. For **LLaVa (LURE)**, the input during inference comprises the original question from POPE, the image, and the caption that has been modified by LURE.

---

> > ### Author Response · Authors · 2023-11-23
> > **We would like to hear additional feedback from reviewer ZMPq**
> >
> > Dear Reviewer ZMPq,
> >
> > Thank you once again for your valuable feedback! As the discussion period is ending soon, we would like to inquire if you have any further feedback regarding our original and additional responses. If you have any questions or suggestions based on our response, we would be delighted to discuss them.

---

> > > ### Comment · Reviewer_ZMPq · 2023-12-03
> > >
> > > Thanks to the authors for the responses. However, the reviewer believes the comparison between the proposed method and the baselines on VQA benchmarks is **not fair**. A fairer baseline should be the comparison between the proposed method with LVLMs + uncorrected caption as context for VQA. Also, it seems the proposed method is still applied to the caption generated by LVLMs.
> > >
> > > The explanations for some points solve some concerns from the reviewer. The experiment results on more caption data and more evaluation about the caption validates the method.
> > >
> > > Overall, it is a meaningful study for detailed caption tasks. The reviewer will raise the score to 6 (somewhere close to 5.5). However, the reviewer suggests a more fair comparison of VQA datasets (which could show the usefulness of better captions).

---

### Official Review · Reviewer_SdU6 · 2023-10-31

**Soundness:** 3 good
**Presentation:** 3 good
**Contribution:** 3 good
**Rating:** 8
**Confidence:** 3

**Summary:**

This paper proposes LURE, a post-hoc approach to reduce object hallucination in large vision-language models (LVLMs). LURE is grounded in a statistical analysis revealing co-occurrence, uncertainty, and object position as key factors causing hallucination. Experiments show LURE outperforms prior methods in reducing hallucination across multiple LVLMs according to general metrics, GPT evaluation, and human evaluation.

**Strengths:**

1. The paper focus on an important problem, object hallucinations in large vision-language models.
2. It spots three key factors of the object hallucinations, the co-occurrence, uncertainty, and object positions.
3. The paper proposes a new post-hoc method to reduce object hallucinations of LVLMs. Extensive experiments verify the effectiveness of proposed methods.

**Weaknesses:**

1.The proposed method helps improve performance on object hallucinations. However, there is a concern that it may harm performance on other metrics like creativity and completeness of captions. It seems to replace detailed words with coarse words as shown in Fig 8.
2.It is unclear if the removed objects are truly hallucinated or if it wrongly removes some non-hallucinated objects. A new metric to quantify this would be helpful.

**Questions:**

1.Do the authors think image captioning metrics are good metrics for LVLMs? The BLEU scores seem low compared to image captioning models. Some important metrics like METEOR, ROUGE, CIDER, and SPICE are missing in Table 10.
2.Why were co-occurrence, uncertainty, and object positions identified as the three key factors for object hallucinations? Were other factors investigated?

---

> ### Author Response · Authors · 2023-11-19
> **Response to Reviewer SdU6 (1/2)**
>
> Thank you for your constructive comments and suggestions. We have revised our paper according to your comments. We respond to your questions below and would appreciate it if you could let us know if our response addresses your concerns.
>
>
> > **Q1**: The proposed method helps improve performance on object hallucinations. However, there is a concern that it may harm performance on other metrics like creativity and completeness of captions. It seems to replace detailed words with coarse words as shown in Fig 8. 2.It is unclear if the removed objects are truly hallucinated or if it wrongly removes some non-hallucinated objects. A new metric to quantify this would be helpful.
>
> **A1**: To delve deeper into the issue of creativity and completeness in captions, we conduct experiments to analyze the variations in the diversity of descriptions generated by different models before and after applying LURE. Our primary focus was on assessing changes in description length and the reduction in the proportion of correctly identified objects, as well as the decrease in hallucinated objects when LURE was introduced. We have presented the findings in Table R1.
>
> The results of our study reveal that the incorporation of LURE leads to a significant reduction in hallucinated objects, averaging around 56%, while only slightly affecting the number of correctly identified objects, with an average decrease of approximately 1.6%. This noteworthy outcome can be attributed to the fact that LURE encourages the model to reevaluate these potentially hallucinatory objects, either removing or replacing them. This approach substantially enhances performance and reduces hallucination. Moreover, this advantage is evident in the average length of descriptions before and after applying LURE. It is clear that LURE has only a minor impact on the length of descriptions, indicating its effectiveness in preserving the utility and diversity of the responses.
>
> In summary, the use of LURE achieves a balance between response accuracy and utility in LVLMs. We have included these newly obtained results in Appendix C.2  of the updated paper.
>
> **Table R1**: Analysis of correctness and usefulness before and after applying LURE.
>
> |                           | MiniGPT-4 | LLaVa  | MMGPT  | LLaMA-Adapter | mPLUG-Owl | InstructBLIP |
> |---------------------------|-----------|--------|--------|---------------|-----------|--------------|
> | Reduction of correct objects (%)      | 0.68     | 2.15  | 1.42  | 1.61         | 1.13     | 2.72        |
> | Reduction of hallucinated object (%) | 41.13     | 56.51  | 69.36  | 52.19         | 61.88     | 54.96        |
> | Average description length (before)   | 67.08     | 102.8  | 63.18  | 94.27         | 110.1     | 95.63        |
> | Average description length (after)    | 56.63     | 96.39  | 57.24  | 93.44         | 99.15     | 92.27        |
>
> ---
>
> > **Q2**: Do the authors think image captioning metrics are good metrics for LVLMs? The BLEU scores seem low compared to image captioning models. Some important metrics like METEOR, ROUGE, CIDER, and SPICE are missing in Table 10.
>
> **A2**: We conduct additional experiments to evaluate performance using METEOR, CIDER, and SPICE, while ROUGE has already been evaluated and is presented in Table 10 of Appendix B.3. The results are detailed in Table R2, demonstrating improvements across various metrics when LURE is applied. We’ve included these results in Table 11 of Appendix B.3  in the updated paper.
>
> However, it is worth noting that while LURE can enhance image captioning metrics in most scenarios, these metrics may not provide a sufficiently accurate evaluation of hallucinations in LVLMs. This limitation arises from the metrics' emphasis on measuring the similarity between generated captions and groundtruth. In the case of fine-grained and detailed captions, these metrics may not accurately reflect the level of object accuracy in the descriptions. This issue occurs because increased object accuracy in fine-grained content may not substantially impact caption similarity.
>
> **Table R2**: Performance with additional metrics.
>
> | Models             |         | METEOR | CIDER | SPICE |
> |--------------------|---------|--------|-------|-------|
> |                    | Original| 28.7   | 0.53  | 17.5  |
> | mPLUG-Owl          | **LURE** | **36.7** | **0.66** | **18.9** |
> |                    | Original| 37.7   | 0.61  | 22.6  |
> | LLaVa              | **LURE** | **43.9** | **0.67** | **31.4** |
> |                    | Original| 27.6   | 0.59  | 21.8  |
> | LLaMA-Adapter      | **LURE** | **33.4** | **0.63** | **29.2** |
> |                    | Original| 22.0   | 0.51  | 17.9  |
> | MiniGPT-4          | **LURE** | **25.6** | **0.55** | **26.4** |
> |                    | Original| 24.3   | 0.56  | 18.9  |
> | MMGPT              | **LURE** | **26.8** | **0.61** | **20.1** |
> |                    | Original| 26.5   | 0.62  | 18.5  |
> | InstructBLIP       | **LURE** | **30.3** | **0.72** | **19.6** |

---

> ### Author Response · Authors · 2023-11-19
> **Response to Reviewer SdU6 (2/2)**
>
> >**Q3**: Why were co-occurrence, uncertainty, and object positions identified as the three key factors for object hallucinations? Were other factors investigated?
>
> **A3**: The selection of these three factors is based on their broad applicability, as they can apply to various types of images. Apart from these factors, we did explore other factors that may contribute to hallucinations. For instance, our investigations revealed that LVLMs are susceptible to hallucinations when confronted with (1) sketch images or sketch objects and (2) composite images. In the case of composite images, combining multiple semantically similar images into a grid can trigger hallucinations in LVLMs. However, it's important to note that these findings are more specific and are primarily associated with certain types of images. Consequently, we have not included them to design LURE.

---

> ### Comment · Reviewer_SdU6 · 2023-11-23
> **Thanks for the responses of the authors.**
>
> Thanks to the rebuttal of the reviewers. I like the paper as it is inspirational on reducing hallucinations in VLLM.
>
> I will keep my score.

---

### Official Review · Reviewer_4L8t · 2023-11-01

**Soundness:** 3 good
**Presentation:** 3 good
**Contribution:** 2 fair
**Rating:** 6
**Confidence:** 4

**Summary:**

The paper proposes a simple algorithm, LVLM Hallucination Revisor (LURE), to post-hoc rectify object hallucination in LVLMs. Their reported results demonstrate that LURE can significantly reduce object hallucination under general object hallucination evaluation metrics.

**Strengths:**

- The paper compares hallucinatory and non-hallucinatory captions from three critical viewpoints, including co-occurrence, uncertainty, and object position. This viewpoint though simple, is instructive.
- LURE is a lightweight and effective post-hoc method, which achieves reasonable performance on six open-source LVLMs.
- LURE consistently improves its performance compared to the original description, which shows its robustness under different backbones.

**Weaknesses:**

- The authors' introduction of their training dataset appears to be insufficiently detailed in certain areas. Firstly, the dataset's composition is not entirely clear, especially concerning the proportion of hallucinated content within it. Secondly, during the training of the hallucination revisor, they used 5,000 image-text pairs from the LLaVA-150k dataset randomly. However, the study does not provide adequate experimental backing to validate the adequacy of this sample size for their objectives. It would be beneficial if the authors could offer a more comprehensive description of their dataset, and ideally make it open-sourced. Such an act could serve as an added contribution to their work. Furthermore, the LURE methodology, as presented, comes across as somewhat straightforward, lacking a distinctive innovative edge.
- LURE is designed as a post-hoc solution aimed at addressing object hallucination; however, it doesn't directly confront the underlying causes of the issue. A more direct challenge would be formulating strategies for guiding the LVLM to produce answers with reduced hallucination tendencies.
- Lack of comparing LURE's performance on the fine-grained caption and concise caption. Intuitively, the problem of hallucination would be more common in fine-grained captions.

**Questions:**

- The effect of object position on object hallucination is not clear. I am still confused why hallucination occurs in the latter part of the descriptions. Is it possible to fundamentally reduce LVLM hallucination from this perspective.

**Details Of Ethics Concerns:**

N/A.

---

> ### Author Response · Authors · 2023-11-19
> **Response to Reviewer 4L8t (1/3)**
>
> Thank you for reviewing our paper and for your valuable feedback. Below, we address your concerns point by point and we’ve revised our paper according to your suggestions. We would appreciate it if you could let us know whether your concerns are addressed by our response.
>
> > **Q1**: The dataset's composition is not entirely clear, especially concerning the proportion of hallucinated content within it. Secondly, during the training of the hallucination revisor, they used 5,000 image-text pairs from the LLaVA-150k dataset randomly. However, the study does not provide adequate experimental backing to validate the adequacy of this sample size for their objectives. It would be beneficial if the authors could offer a more comprehensive description of their dataset, and ideally make it open-sourced. Such an act could serve as an added contribution to their work. Furthermore, the LURE methodology, as presented, comes across as somewhat straightforward, lacking a distinctive innovative edge.
>
> **A1**: Our data construction process involves the creation of examples composed of three key elements: an image, a groundtruth description, and a hallucinatory description. The groundtruth descriptions are sourced from the LLaVA-150k dataset, while the hallucinatory descriptions are generated using an analysis based on three factors derived from real descriptions. This construction process necessitates the use of GPT's API and manual screening. Due to budget considerations, we have currently assembled a dataset comprising 5,000 training data samples. Our experimental results indicate that this dataset size is sufficient for training an effective hallucination revisor.
>
> Furthermore, the groundtruth descriptions in our dataset underwent meticulous manual screening, resulting in very few instances of hallucination. To assess the quality of the groundtruth descriptions, we also calculated the CHAIR metric for both the groundtruth and hallucinatory descriptions in the dataset. It's worth noting that CHAIR, designed to detect 80 specific objects, may not provide entirely accurate results in this context. Nevertheless, the results on both metrics are as follows:
>
> - Groundtruth descriptions: CHAIRs: 3.4, CHAIRi: 0.4
> - Hallucinatory descriptions: CHAIRs: 41.7, CHAIRi: 7.0
>
> We have also made the dataset publicly available. You can access the dataset via the following anonymous link: https://anonymous.4open.science/r/hallucination5k-2A20/hallucination5k_train.jsonl
>
> ---
>
> > **Q2**: Furthermore, the LURE methodology, as presented, comes across as somewhat straightforward, lacking a distinctive innovative edge.
>
> **A2**: We agree that LURE is a seemingly simple method, but its effectiveness in rectifying object hallucinations in LVLMs is demonstrated by our empirical results. The design of LURE, however, is not as straightforward. It stems from our comprehensive and rigorous statistical analysis of the reasons behind LVLM hallucinations.

---

> ### Author Response · Authors · 2023-11-19
> **Response to Reviewer 4L8t (2/3)**
>
> > **Q3**: LURE is designed as a post-hoc solution aimed at addressing object hallucination; however, it doesn't directly confront the underlying causes of the issue. A more direct challenge would be formulating strategies for guiding the LVLM to produce answers with reduced hallucination tendencies.
>
> **A3**: LURE, while being a post-hoc solution, proves to be both promising and effective in mitigating object hallucinations in LVLMs. Its key advantage as a post-hoc rectification method lies in its compatibility—it can seamlessly integrate with multiple LVLM architectures as a plug-and-play module, significantly enhancing response quality while minimizing hallucination. Importantly, it achieves these improvements without requiring additional fine-tuning or updates of pre-trained LVLMs, and it doesn't necessitate access to the private data corpus used in the pre-training of commercial large models.
>
> Furthermore, to strengthen the effectiveness of LURE, we conduct additional experiments on other widely recognized benchmarks, specifically MME [Fu et al., 2023] and POPE [Li et al., 2023] (detailed experimental setups can be found in Appendix B.5). The results, presented in Tables R1 and R2, consistently demonstrate that the incorporation of LURE significantly reduces hallucination in both the POPE and MME datasets, further reinforcing its effectiveness. Detailed results and in-depth analysis are available in Appendix B.5 of the updated paper.
>
> Nevertheless, it's equally important to devise strategies that guide LVLMs to produce answers with reduced hallucination tendencies. This direction holds great promise, and we intend to investigate this problem in the future.
>
> **Table R1**:  POPE results of LLaVa on MSCOCO, A-OKVQA, and GQA.
>
> | Dataset    | Model            | Evaluation Setting | Accuracy | Precision | Recall | F1 Score | Yes (%) |
> |------------|------------------|--------------------|----------|-----------|--------|----------|---------|
> | MSCOCO     |                  | Random             | 54.43    | 52.32     | 99.80  | 68.65    | 95.37   |
> |            | LLaVa (Original) | Popular            | 52.43    | 51.25     | 99.80  | 67.72    | 97.37   |
> |            |                  | Adversarial        | 50.77    | 50.39     | 99.87  | 66.98    | 99.10   |
> |            |                  | Random             |**86.33**    | **89.44**     | 82.40  | **85.77**    | 46.07   |
> |            | **LLaVa (LURE)**     | Popular            | **80.30**    | **79.00**     | 82.53  | **80.73**    | 52.23   |
> |            |                  | Adversarial        | **77.17**    | **74.33**     | 83.00  | **78.43**    | 55.83   |
> | A-OKVQA    |                  | Random             | 50.16    | 50.08     | 99.53  | 66.64    | 99.37   |
> |            | LLaVa (Original) | Popular            | 50.03    | 50.02     | 99.67  | 66.61    | 99.63   |
> |            |                  | Adversarial        | 50.13    | 50.07     | 99.67  | 66.65    | 99.53   |
> |            |                  | Random             | **83.70**    | **84.32**     | 82.80  | **83.55**    | 49.10   |
> |            | **LLaVa (LURE)**     | Popular            | **78.00**    | **75.86**     | 82.13  | **78.87**    | 54.13   |
> |            |                  | Adversarial        | **69.93**    | **65.72**     | 83.33  | **73.49**    | 63.40   |
> | GQA        |                  | Random             | 50.17    | 50.08     | 99.20  | 66.56    | 99.03   |
> |            | LLaVa (Original) | Popular            | 50.03    | 50.02     | 99.47  | 66.56    | 99.43   |
> |            |                  | Adversarial        | 49.77    | 49.88     | 99.20  | 66.38    | 99.43   |
> |            |                  | Random             | **83.32**    | **84.22**     | 82.47  | **83.25**    | 49.15   |
> |            | **LLaVa (LURE)**     | Popular            | **80.85**    | **80.09**     | 82.47  | **81.20**    | 51.62   |
> |            |                  | Adversarial        | **78.74**    | **76.67**     | 82.77  | **79.58**    | 54.03   |
>
>
>
> **Table R2**: Hallucinatory performance of MME  before and after the correction by LURE. Since we found that TN (True Negatives) and FN (False Negatives) are both zero in the MME dataset, the values of accuracy and recall are the same.
> | Models         | Version | Accuracy | Recall | F1 Score |
> |----------------|---------|----------|--------|----------|
> | LLaVa      | Original| 90.0     | 90.0   | 94.7     |
> |                | **LURE (ours)**    | **93.3** | **93.3** | **96.6** |
> | MiniGPT-4  | Original| 93.8     | 93.8   | 96.8     |
> |                | **LURE (ours)**    | **96.7** | **96.7** | **98.3** |
> | Mplug-Owl  | Original| 86.7     | 86.7   | 92.6     |
> |                | **LURE (ours)**    | **93.5** | **93.5** | **96.7** |

---

> ### Author Response · Authors · 2023-11-19
> **Response to Reviewer 4L8t (3/3)**
>
> > **Q4**: Lack of comparing LURE's performance on the fine-grained caption and concise caption. Intuitively, the problem of hallucination would be more common in fine-grained captions.
>
> **A4**:  In our initial experiments, we employed detailed and fine-grained captions, obtained by prompting "Generate a short caption of this image". To further explore the effectiveness of LURE with concise captions, we conduct additional experiments, the results of which are presented in Table R3. The concise descriptions are generated using the prompt 'Generate a short caption of the image.' Our findings indicate that LURE remains effective in reducing object hallucinations even with shorter captions, thus reinforcing its capability in mitigating such issues. We’ve included these results in Appendix C.3 of the updated paper.
>
> **Table R3**: Performance of LURE on concise captions generated by four best-performing LVLMs.
>
> |                    | MiniGPT-4 | MiniGPT-4 | LLaVa  | LLaVa  | LLaMA-Adapter | LLaMA-Adapter | mPLUG-Owl | mPLUG-Owl |
> |--------------------|-----------|-----------|--------|--------|---------------|---------------|-----------|-----------|
> |                    | $C_S$ ↓   | $C_I$ ↓   | $C_S$ ↓| $C_I$ ↓| $C_S$ ↓       | $C_I$ ↓       | $C_S$ ↓   | $C_I$ ↓   |
> | Original           | 18.4      | 7.6       | 33.3   | 10.5   | 23.6          | 8.4           | 14.6      | 7.1       |
> | **LURE (ours)**    | **10.6**      | **3.1**       | **21.2**   | **5.3**    | **20.2**          | **5.4**           | **13.1**      | **3.7**       |
>
>
>
> ---
>
> > **Q5**: The effect of object position on object hallucination is not clear. I am still confused why hallucination occurs in the latter part of the descriptions. Is it possible to fundamentally reduce LVLM hallucination from this perspective?
>
> **A5**: One potential reason for object hallucination occurring later in the generated descriptions is highly related to the 'snowballing effect' [Zhang et al., 2023]. This phenomenon occurs when object hallucinations in the early part of generated descriptions lead to a cascade of accumulated hallucinations in the latter portions.
>
> In LURE, we actively encourage the hallucination revisor to ``reconsider” objects in the latter portion of the descriptions, with the aim of mitigating hallucinations. The results presented in Table 15 (updated paper) provide evidence that LURE is effective at reducing hallucinations in the latter part of the generated descriptions.
>
> In future research, to further resolve this issue, an interesting direction to explore would be the combination of multi-modal retrieval techniques with sentence-by-sentence hallucination rectification. This approach could offer a promising direction for addressing the 'snowball' issue. Such research would be considered as part of our future work.
>
> ---
>
> **Reference**
>
> [Zhang et al., 2023] Zhang M, Press O, Merrill W, et al. How language model hallucinations can snowball. arXiv preprint arXiv:2305.13534, 2023.
>
> [Fu et al., 2023] Fu, Chaoyou, et al. "MME: A Comprehensive Evaluation Benchmark for Multimodal Large Language Models." arXiv preprint arXiv:2306.13394 (2023).
>
> [Li et al., 2023] Li, Yifan, et al. "Evaluating object hallucination in large vision-language models." arXiv preprint arXiv:2305.10355 (2023).

---

### Official Review · Reviewer_4tjW · 2023-11-01

**Soundness:** 2 fair
**Presentation:** 3 good
**Contribution:** 3 good
**Rating:** 5
**Confidence:** 4

**Summary:**

This paper finds three key factors related to object hallucination: co-occurrence, uncertainty, and object position. Based on this, the authors propose LVLM Hallucination Revisor (LURE) to rectify the object hallucination issue in LVLMs. LURE takes text descriptions as input and outputs refined ones. The authors collect a hallucinatory dataset using GPT-3.5 and thereby train the LURE. The experiments evaluate LURE on existing open-source LVLMs and results demonstrate LURE's effectiveness.

**Strengths:**

1. This paper proposes a framework to address the hallucination of LVLMs, by identifying key factors and training a revisor correspondingly.
2. This paper presents the technical details clearly. Rigorous theoretical derivations are provided as well.
3. This paper conducts extensive experiments and shows the quantitative improvements of LURE.

**Weaknesses:**

1. The definition of positioning score is not intuitive. Have the authors analyzed the position score distribution under different description lengths? If shorter descriptions yield lower position hallucination, would generating multiple short descriptions and combining them result in a high-quality description?
2. Lack of results of other popular benchmarks. This paper only reports the performance on the COCO 2014 test dataset, which is small and may be biased. There is no result about the performance on other popular benchmarks for LVLMs, such as MMBench, MME, POPE, SEED, MM-Vet, etc. Will the performances be better or worse on these benchmarks?
3. Lack of an analysis of the complexity and usefulness of the responses. There is a tradeoff between the correctness and complexity of the responses. Directly removing the hallucination context may improve the correctness but reduce the diversity and complexity. An analysis regarding this concern is important for a comprehensive understanding of the impact of the proposed method.

**Questions:**

The results of Figure 1(c). Is this quantity of images (just 200) sufficient to consolidate the distribution statistics? And even if a sufficient number of samples are provided in a specific domain, can this conclusion be generalized to the distribution of other datasets or benchmarks?

---

> ### Author Response · Authors · 2023-11-19
> **Response to Reviewer 4tjW (1/3)**
>
> Thank you for your valuable feedback to help us improve our paper. We have revised our paper based on your feedback. We detail our response below and please kindly let us know if our response addresses your concerns.
>
> > **Q1**: Comprehensive analysis of object position and hallucination, including subquestion “the definition of positioning score is not intuitive. Have the authors analyzed the position score distribution under different description lengths?” and “The results of Figure 1(c). Is this quantity of images (just 200) sufficient to consolidate the distribution statistics? And even if a sufficient number of samples are provided in a specific domain, can this conclusion be generalized to the distribution of other datasets or benchmarks?”
>
> **A1**: To gain a deeper understanding of the impact of object position on hallucinations, we extend our analysis beyond the existing evaluation presented in Figure 1(c). This extended analysis encompasses the following evaluations:
>
> - **Evaluation with More Examples**: In the first phase of our evaluation, we re-assess the distribution of hallucinatory objects concerning their positions using a larger dataset comprising 5,000 examples from the COCO dataset. The results are detailed in Figure 5(a) within Appendix C.1.1 of our updated paper.
>
> - **Evaluation on short descriptions**: Similarly, in the second phase, we evaluate the distribution of hallucinatory objects concerning their positions within short descriptions generated by models such as OFA, BLIP2, etc., using the same 5,000 data points as in the first evaluation. These findings are illustrated in Figure 5(b) of Appendix C.1.1 in our updated paper.
> - **Evaluation on other datasets**: In the third phase, we explore the connection between the distribution of hallucinatory objects and their positions in datasets such as CC dataset. For this evaluation, descriptions are manually annotated to identify hallucinated objects, and the results are reported in Figure 5(c) of Appendix C.1.1.
>
> Across all evaluations, our findings consistently indicate that high-density areas of hallucinatory objects predominantly appear towards the end of the sequence, regardless of the length of the descriptions. This further reinforces our original conclusions. Furthermore, it is worth noting that generating shorter descriptions does not yield lower position hallucination. Therefore, simply generating multiple short descriptions and combining them may not necessarily lead to higher-quality descriptions.

---

> ### Author Response · Authors · 2023-11-19
> **Response to Reviewer 4tjW (2/3)**
>
> > **Q2**: Lack of results of other popular benchmarks. This paper only reports the performance of the COCO 2014 test dataset, which is small and may be biased. There is no result about the performance on other popular benchmarks for LVLMs, such as MMBench, MME, POPE, SEED, MM-Vet, etc. Will the performances be better or worse on these benchmarks?
>
> **A2**: We conduct additional experiments for LURE on other popular benchmarks, specifically MME and POPE, as they are the most suitable datasets for evaluating hallucinations (refer to the comprehensive experimental setups in Appendix B.5). The results are presented in Tables R1 and R2, where we observe a noteworthy reduction in hallucination with the introduction of LURE in both the POPE and MME datasets. These results not only underscore the effectiveness of LURE but also provide additional support for the conclusions drawn in our original paper. We have included these new findings and an in-depth analysis in Appendix B.5 of the updated paper.
>
> **Table R1**:  POPE results of LLaVa on MSCOCO, A-OKVQA, and GQA.
>
> | Dataset    | Model            | Evaluation Setting | Accuracy | Precision | Recall | F1 Score | Yes (%) |
> |------------|------------------|--------------------|----------|-----------|--------|----------|---------|
> | MSCOCO     |                  | Random             | 54.43    | 52.32     | 99.80  | 68.65    | 95.37   |
> |            | LLaVa (Original) | Popular            | 52.43    | 51.25     | 99.80  | 67.72    | 97.37   |
> |            |                  | Adversarial        | 50.77    | 50.39     | 99.87  | 66.98    | 99.10   |
> |            |                  | Random             |**86.33**    | **89.44**     | 82.40  | **85.77**    | 46.07   |
> |            | **LLaVa (LURE)**     | Popular            | **80.30**    | **79.00**     | 82.53  | **80.73**    | 52.23   |
> |            |                  | Adversarial        | **77.17**    | **74.33**     | 83.00  | **78.43**    | 55.83   |
> | A-OKVQA    |                  | Random             | 50.16    | 50.08     | 99.53  | 66.64    | 99.37   |
> |            | LLaVa (Original) | Popular            | 50.03    | 50.02     | 99.67  | 66.61    | 99.63   |
> |            |                  | Adversarial        | 50.13    | 50.07     | 99.67  | 66.65    | 99.53   |
> |            |                  | Random             | **83.70**    | **84.32**     | 82.80  | **83.55**    | 49.10   |
> |            | **LLaVa (LURE)**     | Popular            | **78.00**    | **75.86**     | 82.13  | **78.87**    | 54.13   |
> |            |                  | Adversarial        | **69.93**    | **65.72**     | 83.33  | **73.49**    | 63.40   |
> | GQA        |                  | Random             | 50.17    | 50.08     | 99.20  | 66.56    | 99.03   |
> |            | LLaVa (Original) | Popular            | 50.03    | 50.02     | 99.47  | 66.56    | 99.43   |
> |            |                  | Adversarial        | 49.77    | 49.88     | 99.20  | 66.38    | 99.43   |
> |            |                  | Random             | **83.32**    | **84.22**     | 82.47  | **83.25**    | 49.15   |
> |            | **LLaVa (LURE)**     | Popular            | **80.85**    | **80.09**     | 82.47  | **81.20**    | 51.62   |
> |            |                  | Adversarial        | **78.74**    | **76.67**     | 82.77  | **79.58**    | 54.03   |
>
>
>
> **Table R2**: Hallucinatory performance of MME  before and after the correction by LURE. Since we found that TN (True Negatives) and FN (False Negatives) are both zero in the MME dataset, the values of accuracy and recall are the same.
> | Models         | Version | Accuracy | Recall | F1 Score |
> |----------------|---------|----------|--------|----------|
> | LLaVa      | Original| 90.0     | 90.0   | 94.7     |
> |                | **LURE (ours)**    | **93.3** | **93.3** | **96.6** |
> | MiniGPT-4  | Original| 93.8     | 93.8   | 96.8     |
> |                | **LURE (ours)**    | **96.7** | **96.7** | **98.3** |
> | Mplug-Owl  | Original| 86.7     | 86.7   | 92.6     |
> |                | **LURE (ours)**    | **93.5** | **93.5** | **96.7** |

---

> > ### Author Response · Authors · 2023-11-19
> > **Response to Reviewer 4tjW (3/3)**
> >
> > > **Q3**: Lack of an analysis of the complexity and usefulness of the responses. There is a tradeoff between the correctness and complexity of the responses. Directly removing the hallucination context may improve the correctness but reduce the diversity and complexity. An analysis regarding this concern is important for a comprehensive understanding of the impact of the proposed method.
> >
> > **A3**: We conduct additional analyses to examine the impact of LURE on the diversity and completeness of descriptions generated by various models before and after applying LURE. Our primary focus is on several key aspects: changes in description length, reduction in the proportion of correctly identified objects, and reduction in hallucinatory objects after applying LURE. The detailed results on six LVLMs are presented in Table R3.
> >
> > Our findings reveal that the incorporation of LURE leads to a significant reduction in hallucinatory objects, averaging around 56%, while only slightly affecting the presence of correctly identified objects, with an average decrease of approximately 1.6%. This noteworthy outcome can be attributed to the fact that LURE doesn't merely eliminate potentially hallucinatory objects; it actively encourages the model to reconsider and either remove or replace such objects. This approach significantly enhances model performance and reduces hallucination. Furthermore, the positive effect of LURE is evident in the average length of the generated descriptions. Applying LURE results in only minor changes to the description length, indicating its effectiveness in preserving the utility and diversity of the generated responses.
> >
> > In summary, the use of LURE achieves a balance between the correctness and usefulness of responses in LVLMs. We have updated our paper to incorporate these results and discussions in Appendix C.2.
> >
> >
> > **Table R3**: Analysis of correctness and usefulness before and after applying LURE.
> >
> > |                           | MiniGPT-4 | LLaVa  | MMGPT  | LLaMA-Adapter | mPLUG-Owl | InstructBLIP |
> > |---------------------------|-----------|--------|--------|---------------|-----------|--------------|
> > | Reduction of correct objects (%)      | 0.68     | 2.15  | 1.42  | 1.61         | 1.13     | 2.72        |
> > | Reduction of hallucinated object (%) | 41.13     | 56.51  | 69.36  | 52.19         | 61.88     | 54.96        |
> > | Average description length (before)   | 67.08     | 102.8  | 63.18  | 94.27         | 110.1     | 95.63        |
> > | Average description length (after)    | 56.63     | 96.39  | 57.24  | 93.44         | 99.15     | 92.27        |

---

> > > ### Author Response · Authors · 2023-11-21
> > > **We would like to hear back from reviewer 4tjW**
> > >
> > > Dear reviewer 4tjW,
> > >
> > > We would like to follow up to see if the response addresses your concerns or if you have any further questions. We would really appreciate the opportunity to discuss this further if our response has not already addressed your concerns. Thank you again!

---

### Author Response · Authors · 2023-11-19
**Summary of Paper Revision**

We sincerely appreciate all reviewers for their insightful and constructive feedback. According to these comments, we have improved the paper (new pdf uploaded) and highlighted the main changes with blue text. Below, we summarize all changes:

1. We have included an analysis of experiments conducted on two additional benchmarks (POPE and MME) and two additional datasets (ImageNet and Conceptual Captions). (Reviewers 4tjW, 4L8t, and ZMPq)

2. We expanded the analysis of the relationship between object position and hallucination in Figure 1(c). The results are presented in Figure 5 in Appendix C.1.1. (Reviewer 4tjW)

3. We added an analysis of object hallucinations in short captions and their effects before and after applying LURE. This analysis and its results can be found in Appendix C.3. (Reviewer 4L8t)

4. We analyzed changes in description diversity and completeness before and after applying LURE. The specific analysis can be found in Appendix C.2. (Reviewers 4tjW and SdU6)

5. We have included additional metrics in Appendix B.3.1, and the results are presented in Table 11. (Reviewer SdU6)

6. Due to the inclusion of more results and discussions in the main paper and appendix (Reviewers 4tjW, 4L8t, SdU6, and ZMPq), some appendix indexes have been updated.

---

### Meta-Review · Area_Chair_pzAE · 2023-12-12

**Metareview:**

This paper presents LURE, a novel method to address object hallucination in LVLMs. Specifically, it identifies co-occurrence, uncertainty, and object position as key factors contributing to hallucinations, and proposes a dataset and a revisor model to mitigate these issues. Overall, the reviewers enjoy reading this paper and agree that the proposed LURE provides a simple yet effective method for mitigating object hallucination in LVLMs. There are some concerns raised, including 1) the generalizability of LURE could be limited, 2) the lack of detailed information on the training dataset composition, and 3) more empirical ablations are needed. The authors provide a detailed rebuttal, which, we believe, provides sufficient evidence to reasonably address these concerns. Overall, we recommend accepting this paper.

In the final version, the authors should include these additional clarifications/ablations provided in the rebuttal to enhance the quality of this paper.

**Justification For Why Not Higher Score:**

As mentioned by Reviewer 4L8t, while this paper provides an effective post-hoc solution for mitigating object hallucination, it fails to touch the underlying causes of this issue (e.g., where such hallucinations come from and how we can fix it in training). Therefore we are unable to consider it for a higher recommendation.

**Justification For Why Not Lower Score:**

Overall, this paper presents a simple and effective strategy for mitigating object hallucinations in LVLMs, which deserves a poster presentation at ICLR.

---

### Decision · Program_Chairs · 2024-01-16

Accept (poster)